# Chl1 helicase controls replication fork progression by regulating dNTP pools

Amandine Batté[1,*] ⓘ, Sophie C van der Horst[1,*] ⓘ, Mireille Tittel-Elmer[1,2], Su Ming Sun[1], Sushma Sharma[3] ⓘ, Jolanda van Leeuwen[4] ⓘ, Andrei Chabes[3], Haico van Attikum[1] ⓘ

Eukaryotic cells have evolved a replication stress response that helps to overcome stalled/collapsed replication forks and ensure proper DNA replication. The replication checkpoint protein Mrc1 plays important roles in these processes, although its functional interactions are not fully understood. Here, we show that *MRC1* negatively interacts with *CHL1*, which encodes the helicase protein Chl1, suggesting distinct roles for these factors during the replication stress response. Indeed, whereas Mrc1 is known to facilitate the restart of stalled replication forks, we uncovered that Chl1 controls replication fork rate under replication stress conditions. Chl1 loss leads to increased *RNR1* gene expression and dNTP levels at the onset of S phase likely without activating the DNA damage response. This in turn impairs the formation of RPA-coated ssDNA and subsequent checkpoint activation. Thus, the Chl1 helicase affects RPA-dependent checkpoint activation in response to replication fork arrest by ensuring proper intracellular dNTP levels, thereby controlling replication fork progression under replication stress conditions.

## Introduction

Faithful duplication of the genome by DNA replication in the S phase of the cell cycle is crucial for the maintenance of genomic stability. However, this process can be compromised when replication forks encounter obstacles such as DNA lesions, secondary DNA structures, natural pause sites, or covalent protein-DNA crosslinks, that can lead to replication fork stalling (Zeman & Cimprich, 2014). Stalled forks accumulate single-stranded DNA (ssDNA) that is prone to breakage and the formation of toxic recombination intermediates if they are not properly stabilized and restarted. Fortunately, eukaryotic cells have evolved a replication stress response that protects the integrity of replication forks

(Branzei & Foiani, 2009; Labib & De Piccoli, 2011; Pardo et al, 2017). The DNA replication checkpoint is a major pathway of this surveillance mechanism mediated by the highly conserved Mec1/ATR and Rad53/Chk2 kinases. The sensor kinase Mec1/ATR detects the accumulation of replication protein A (RPA)-coated ssDNA at stalled forks and promotes the phosphorylation of the effector kinase Rad53/Chk2. In *Saccharomyces cerevisiae*, activated Rad53 maintains replication fork integrity and controls inhibition of late origin firing, up-regulation of dNTP pools and activation of DNA damage repair genes (Huang et al, 1998; Lopes et al, 2001; Sogo et al, 2002; Zhao & Rothstein, 2002; Zegerman & Diffley, 2010).

Several factors involved in the protection of arrested replication forks are also involved in sister chromatid cohesion (SCC) (Warren et al, 2004; Xu et al, 2004; Lengronne et al, 2006). Cohesion between sister chromatids ensures the faithful segregation of genetic information into daughter cells and is achieved by the structural maintenance of chromosome (SMC) complex cohesin. This ring-shaped complex is composed of two SMC proteins, Smc1 and Smc3, which are bridged by Scc1/Rad21 and Scc3/SA (Michaelis et al, 1997; Xiong & Gerton, 2010). Cohesin also participates in homologous recombination (Strom et al, 2004; Unal et al, 2004, 2007; Gelot et al, 2016) and associates with replication forks in both yeast and mammals to promote fork restart and telomeric replication (Remeseiro et al, 2012; Tittel-Elmer et al, 2012; Frattini et al, 2017). However, how cohesin and cohesin-associated factors interact with stalled replication forks and promote their stabilization/restart is still poorly understood.

Chl1 is the yeast counterpart of human ChlR1/DDX11, an ATP-dependent DEAH-box DNA helicase that progresses along ssDNA and unwinds DNA in a 5' to 3' directionality (Hirota & Lahti, 2000; Farina et al, 2008). Moreover, Chl1/DDX11 are also auxiliary cohesin factor that promote SCC in yeast and human cells independently of Eco1 (Mayer et al, 2004; Petronczki et al, 2004; Skibbens, 2004; Parish et al, 2006; Farina et al, 2008; Borges et al, 2013). Chl1 does so by acting directly at replication forks, where it facilitates the accrual of the cohesin loader Scc2, regulates the acetylation of Smc3, and

[1]Department of Human Genetics, Leiden University Medical Center, Leiden, Netherlands   [2]Electrical Engineering, Mathematics and Computer Science, Delft University of Technology, Delft, Netherlands   [3]Department of Medical Biochemistry and Biophysics, Umeå University, Umeå, Sweden   [4]Center for Integrative Genomics, Université de Lausanne, Lausanne-Dorigny, Switzerland

Correspondence: h.van.attikum@lumc.nl
Amandine Batté's present address is Center for Integrative Genomics, Université de Lausanne, Lausanne-Dorigny, Switzerland.
*Amandine Batté and Sophie C van der Horst contributed equally to this work.

promotes chromatin accessibility through nascent DNA resection (Rudra & Skibbens, 2013; Samora et al, 2016; Delamarre et al, 2020). Chl1/DDX11 also preserve cell viability after exposure of yeast and human cells to various DNA damaging agents. However, whether this effect can be attributed to Chl1's role in SCC or a role in the protection of damaged replication forks remains unclear (Laha et al, 2006, 2011; Parish et al, 2006; Ogiwara et al, 2007; Shah et al, 2013). Importantly, mutations in *DDX11* are causally linked to the rare cohesinopathy-related disease called Warsaw breakage syndrome, which features both impaired SCC and increased chromosomal breakage (van der Lelij et al, 2010). Moreover, DDX11 helps to bypass G4 structures and sustains checkpoint activation through efficient ssDNA formation and RPA loading (Lerner et al, 2020; Simon et al, 2020; van Schie et al, 2020), suggesting that DDX11 preserves both SCC and genome integrity.

We previously generated a genetic network centered on cohesin (Sun et al, 2020). This network revealed a strong negative genetic interaction between *CHL1* and the replication checkpoint gene *MRC1* under normal conditions as well as under conditions of replication stress, implying a role for Chl1 in the replication stress response. Indeed, while loss of Mrc1 de-stabilizes stalled replication forks (Katou et al, 2003; Szyjka et al, 2005; Tourriere et al, 2005), we found that loss of Chl1's helicase activity regulates the dNTP pools by modulating *RNR1* expression at the onset of S phase. This impairs the formation of RPA-coated ssDNA, leading to defective checkpoint activation. Thus, the Chl1 helicase ensures proper intracellular dNTP levels, thereby controlling RPA-dependent checkpoint activation and replication fork progression under replication stress conditions.

# Results

## Genetic interaction between *CHL1* and *MRC1* implicates *CHL1* in checkpoint control and replication fork progression

To gain more insight into cohesin's mode of action, we recently mapped genetic interactions between 17 cohesin and cohesin-related factors, including *CHL1*, and 1,494 genes involved in various biological processes in the budding yeast *S. cerevisiae* (Sun et al, 2020). Analysis of these interactions revealed that *CHL1* is a major hub in the cohesin network (Fig 1A), showing interactions with genes involved in SCC (*RTS1*, *VIK1*, *DCC1*, *RAD61*, and *CHL4*), chromosome segregation (*BUB1*, *BUB3*, *CTF3*, and *MAD3*) and DNA replication (*POL30* [PCNA], and *POL2* [DNA polymerase ε]). Interestingly, we also identified a strong negative interaction between *CHL1* and *MRC1*, in agreement with previous studies (Xu et al, 2007; Costanzo et al, 2016). Although implicated in SCC (Xu et al, 2004), Mrc1 primarily functions in response to replication perturbations. After replication fork stalling, Mrc1 becomes phosphorylated by Mec1, which promotes the recruitment of Rad53 and activation of the intra-S checkpoint (Alcasabas et al, 2001). Furthermore, Mrc1 has a structural function and is necessary for the stabilization of normal and stalled replication forks (Katou et al, 2003; Szyjka et al, 2005; Tourriere et al, 2005). The interaction between *CHL1* and *MRC1* therefore suggested a role for Chl1 in checkpoint control and/or replication fork maintenance.

To investigate this further, we generated *chl1Δ*, *mrc1Δ*, and *chl1Δ mrc1Δ* strains de novo. Growth of *chl1Δ mrc1Δ* was reduced when

compared with that of *chl1Δ* and *mrc1Δ* alone in unperturbed conditions, which confirmed the negative interaction between *chl1Δ* and *mrc1Δ*, as well as under conditions of replication stress induced by the ribonucleotide reductase inhibitor hydroxyurea (HU) (Fig 1B). Because Mrc1 is implicated in both checkpoint activation and replication fork protection (Alcasabas et al, 2001; Katou et al, 2003; Osborn & Elledge, 2003; Szyjka et al, 2005; Tourriere et al, 2005), we next examined whether loss of *CHL1* is additive with two separation-of-function alleles of *MRC1*. *mrc1^{AQ}* suffers from a checkpoint signaling defect (Osborn & Elledge, 2003), whereas *mrc1^{1-843}* exhibits a slow DNA replication phenotype (Srivatsan et al, 2018). Surprisingly, *chl1Δ* was synthetic sick in combination with both *mrc1^{AQ}* and *mrc1^{1-843}* on HU (Fig 1C), suggesting that Chl1 acts in parallel to Mrc1 both in checkpoint control and replication fork progression.

Chl1 and its human counterpart, DDX11, exhibit DNA-dependent ATPase and DNA helicase activities (Hirota & Lahti, 2000; Farina et al, 2008; Wu et al, 2012). We next asked whether these activities are required for Chl1's role in response to replication stress. To this end, we replaced a conserved lysine in the ATP binding site of Chl1 with arginine (Chl1^{K48R}), as this was shown to abolish helicase activity (Hirota & Lahti, 2000; Farina et al, 2008; Samora et al, 2016). In agreement with a previous report (Samora et al, 2016), expression of WT *CHL1* rescued the HU sensitivity of *chl1Δ*, whereas expression of the *chl1^{K48R}* mutant did not, since these cells were as sensitive to HU as cells carrying the empty vector (EV; Fig 1D). Importantly, expression of the *chl1^{K48R}* mutant, but not WT *CHL1*, neither rescued the HU sensitivity of *chl1Δ mrc1Δ* to the level of *mrc1Δ*, nor that of *chl1Δ mrc1^{AQ}* and *chl1Δ mrc1^{1-843}* to the level of *mrc1^{AQ}* and *mrc1^{1-843}*, respectively (Fig 1E). Altogether, these data suggest that Chl1's helicase activity is required for both checkpoint control and replication fork progression.

## Chl1 associates with stalled replication forks

To assess how Chl1 affects these processes, we first examined whether it is a constituent of the replisome under stress conditions. To this end, DNA polymerase α (Pol α)-HA and Chl1-Myc were immunoprecipitated from extracts of cells that were synchronized in G1 phase and released into HU for 40 min (Fig S1A). Chl1 and Pol α interacted reciprocally, suggesting that Chl1 is part of the replication fork machinery during replication stress. This interaction was conserved in unperturbed conditions (Fig S1B) and likely involves the binding of both proteins to Ctf4 (Samora et al, 2016).

We then determined whether Chl1 associates with stalled replication forks by comparing its localization to that of the DNA polymerase ε (Pol ε) subunit Pol2 at different distances from two early origins (ARS607; ARS305) and a late (ARS501) origin of replication using chromatin immunoprecipitation (ChIP) in combination with quantitative PCR (qPCR) (Fig 2A). Corroborating previous findings (Cobb et al, 2003), we observed that 20 min after release from G1, Pol ε was efficiently recruited to ARS607 and ARS305 and progressed along the chromosome arm at later time points (Fig S1C and D), whereas it was absent from the late firing origin ARS501 (Fig S1C). Importantly, Chl1 was also recruited within 20 min to ARS607 and ARS305, and moved away from these origins at later time points, whereas it was absent from ARS501 (Figs 2B and S1E). In agreement with a previous report (Samora et al, 2016), we also

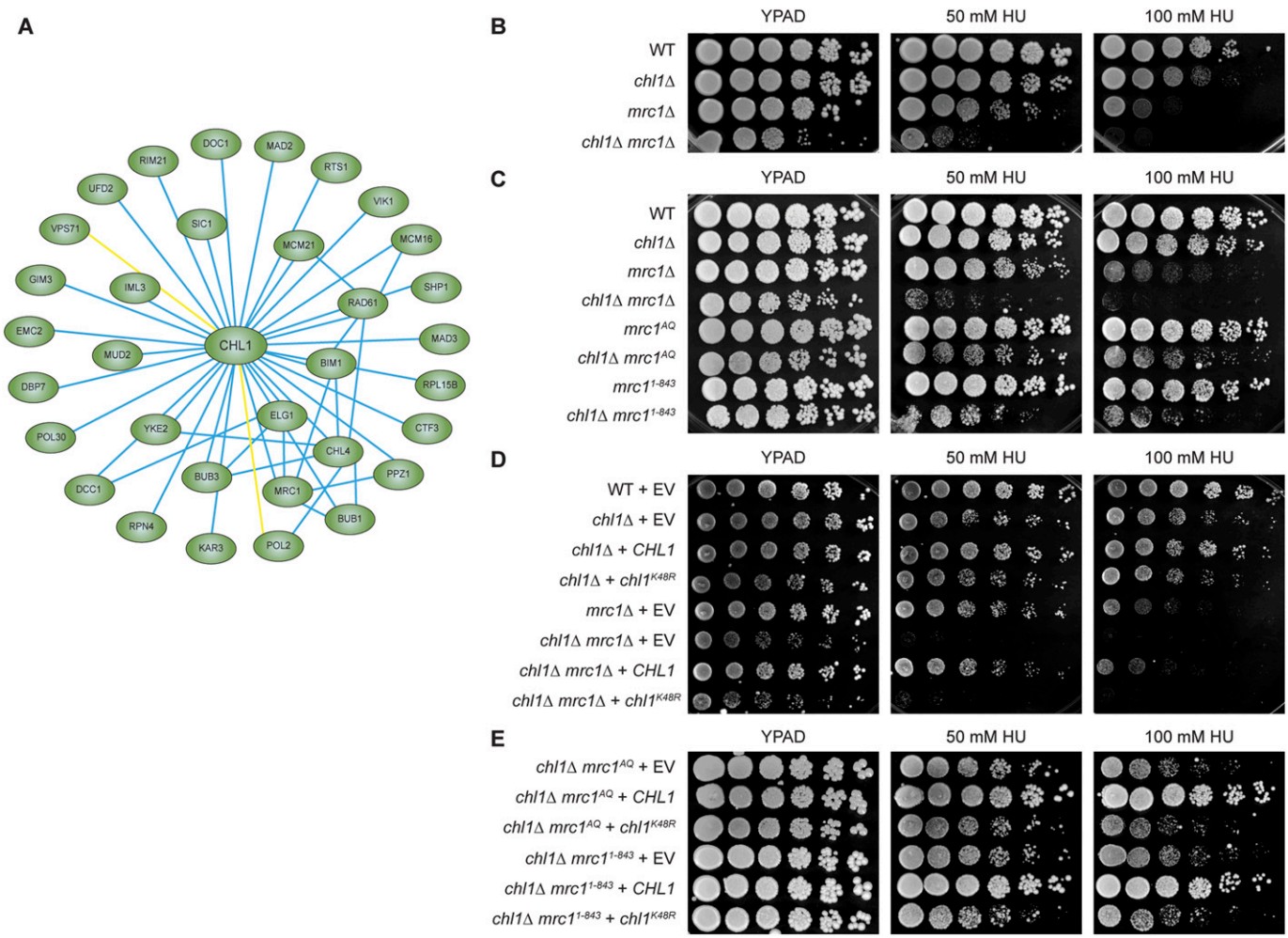

**Figure 1. *CHL1* helicase-dead mutant negatively interacts with both replication- and checkpoint-defective mutants of *MRC1*.**
**(A)** Visualization of *CHL1*'s interaction network. Blue lines represent negative genetic interactions, whereas yellow lines represent positive genetic interactions. **(B)** Spot dilution assay for WT, *chl1Δ*, *mrc1Δ* and *chl1Δ mrc1Δ*. 10-fold serial dilutions of exponentially growing cells were spotted on rich medium (YPAD) without or with HU. **(C)** As in (B), except for WT, *chl1Δ*, *mrc1Δ*, *chl1Δ mrc1Δ*, *mrc1^{AQ}*, *chl1Δ mrc1^{AQ}*, *mrc1^{1-843}* and *chl1Δ mrc1^{1-843}*. **(D)** As in (B), except for WT, *chl1Δ*, *mrc1Δ* and *chl1Δ mrc1Δ* expressing empty vector (EV) (pRS305), *CHL1* (pRS305-*CHL1*) or *chl1^{K48R}* (pRS305-*chl1K48R*). **(E)** Spot dilution assay for *chl1Δ*, *mrc1^{AQ}* and *chl1Δ mrc1^{1-843}* expressing empty vector (EV) (pRS303), *CHL1* (pRS303-*CHL1*) or *chl1^{K48R}* (pRS303-*chl1K48R*). Fivefold serial dilutions of exponentially growing cells were spotted on rich medium (YPAD) without or with HU.

found Chl1 to bind ARS607 and ARS305 in unperturbed conditions in a manner comparable to Pol ε (Fig S1F–I). These data indicate that Chl1 is recruited to replication forks and associates with the replisome both in unperturbed and stressed conditions.

### Chl1-mediated cohesin loading at stalled replication forks does not contribute to fork recovery

Previous studies suggested that Chl1 physically interacts with the cohesin complex at replication forks, and promotes cohesin loading specifically during S phase (Rudra & Skibbens, 2013; Samora et al, 2016). We therefore asked whether Chl1 affects the association of cohesin with stalled replication forks. We found that the cohesin subunit Scc1 was recruited to ARS607 and ARS305 40 min after release of cells from G1 into HU (Figs 2C and S1J), which was 20 min later than Pol ε (Fig S1C), suggesting that at least a

proportion of Scc1 may accumulate on chromatin following passage of the replisome. In addition, Scc1 recruitment was not restricted to stalled replication forks, as it was also observed at the late firing origin ARS501 (Fig 2C), at a convergent intergenic region Conv32W-31C and at two loci in the rDNA (CARL3 and CARL3-N) (Fig S1K). Importantly, deletion of *CHL1* impaired Scc1 accumulation at ARS607 and ARS305 by two to threefold, whereas it was not or only slightly reduced at ARS501, Conv32W-31C, CARL3, and CARL3-N (Figs 2C and S1J and K), suggesting an effect of Chl1 loss on the loading of cohesin at stalled replication forks but not at other loci.

Given that Mrc1 contributes to SCC in a separate pathway to Chl1 (Xu et al, 2007), we next examined whether Chl1 and Mrc1 also act synergistically to promote cohesin loading at stalled replication forks (Figs 2D and S1L). Scc1 binding was only slightly decreased at ARS607 and ARS305 in *mrc1Δ* when compared with WT, suggesting that Mrc1 does not play a substantial role in cohesin loading at

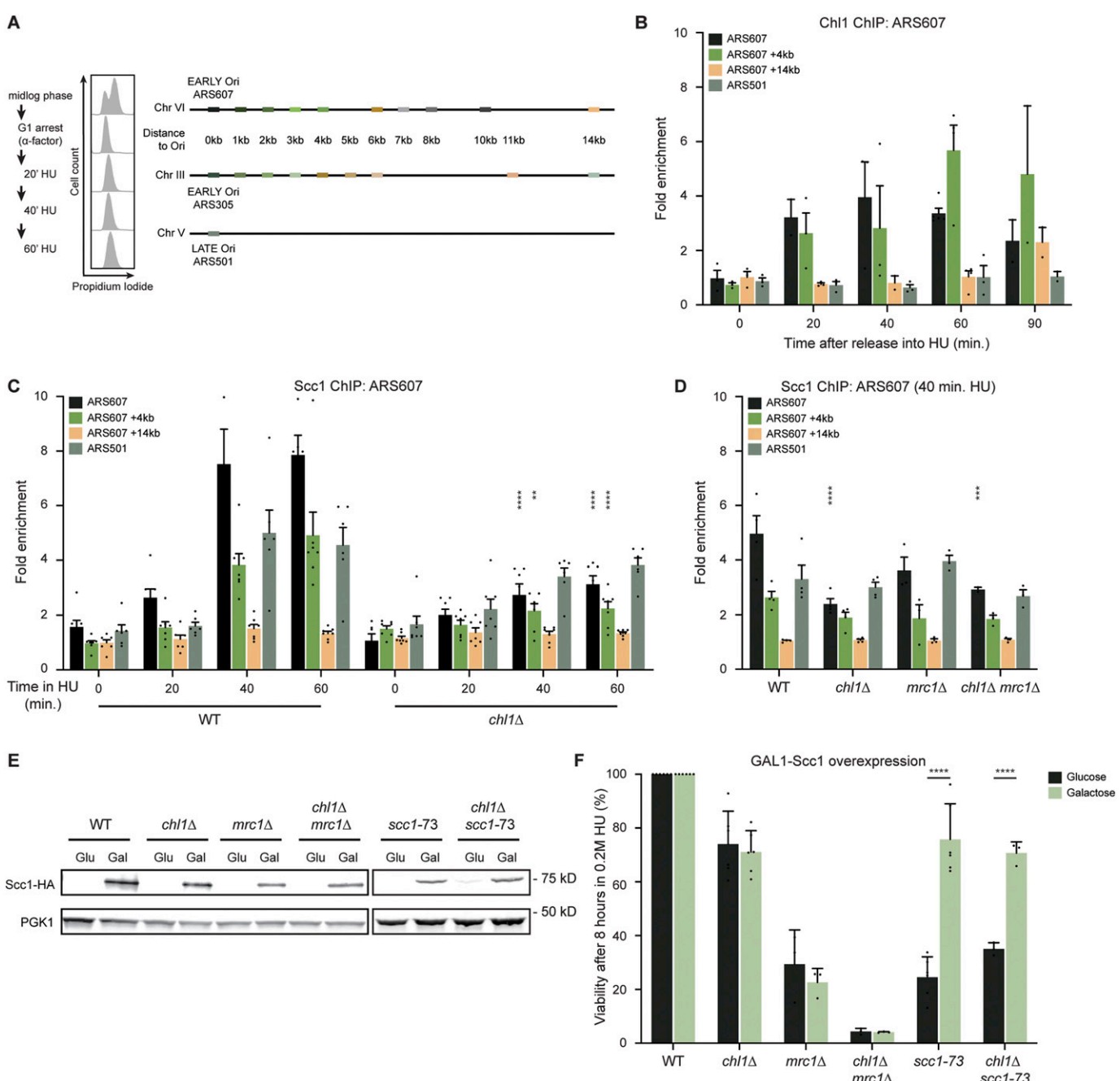

**Figure 2. Chl1 recruits cohesin to stalled replication forks.**
**(A)** Schematic representation of the ChIP approach to monitor replication under stress conditions. Cells were synchronized in G1 phase, released in S phase in the presence of 0.2 M HU, and harvested for ChIP-qPCR at the indicated time points. Colored boxes indicate positions of qPCR primers at the early-firing origins ARS607 and ARS305 and their proximal regions, as well as the late firing origin ARS501. **(B)** Quantification of recruitment of Chl1-Myc at ARS607 by ChIP-qPCR as in (A). Enrichment corresponds to the ratio of the signal after immunoprecipitation (Myc) over beads alone. Error bars represent the standard error of the mean of three independent experiments. **(C)** As in (B), except for quantification of recruitment of Scc1-Myc in WT and chl1Δ. **(D)** As in (C), except in WT, chl1Δ, mrc1Δ and chl1Δ mrc1Δ exposed to HU for 40 min. **(E)** Western blot analysis of Scc1-HA expression after transient exposure of cells to 0.2 M HU and 8 h incubation in medium with galactose (Gal; Scc1-HA overexpression) or glucose (Glu; Scc1-HA repression). **(F)** Survival frequencies observed for WT, chl1Δ, mrc1Δ, chl1Δ mrc1Δ, scc1-73 and chl1Δ scc1-73 from (E). The percentage of survival is normalized to the cell viability at time 0. Error bars represent the SD of three or more independent experiments. Asterisks indicate statistical differences using a t test (*P < 0.05, **P < 0.01, ***P < 0.005, ****P < 0.001).

stalled replication forks. Importantly, Scc1 recruitment was not further diminished in chl1Δ mrc1Δ compared to that in chl1Δ, indicating that Chl1 and Mrc1 do not act in compensatory pathways to load cohesin at stalled forks.

The association of cohesin with stalled replication forks promotes fork restart (Tittel-Elmer et al, 2012; Frattini et al, 2017). To assess the functional relevance of Chl1-dependent cohesin loading at stalled forks, we examined whether overexpression of Scc1 could

rescue the viability of *chl1Δ*, *mrc1Δ* and *chl1Δ mrc1Δ* after transient exposure to HU (Fig 2E). WT cells recovered from the HU exposure, regardless of Scc1 overexpression from a galactose-inducible *GAL1* promoter (Fig 2F). In contrast, the thermo-sensitive *scc1-73* single mutant and the *chl1Δ scc1-73* double mutant showed a drop in viability, which could be restored by Scc1 overexpression at semi-permissive temperature, illustrating the validity of our approach (Tittel-Elmer et al, 2009) and indicating that Scc1 overexpression does not require Chl1 to be effective. In agreement with their inability to resume replication after exposure to HU (Tourriere et al, 2005), *mrc1Δ* cells showed a 70% drop in viability. However, this reduced viability could not be rescued by Scc1 overexpression, consistent with Mrc1 being dispensable for cohesin loading at stalled forks. The viability of *chl1Δ* and *chl1Δ mrc1Δ* was decreased by 20% and 95%, respectively, indicating that Chl1 and Mrc1 have a synergistic effect on the recovery from HU-induced fork stalling. Importantly, Scc1 overexpression did not rescue the survival of *chl1Δ* and *chl1Δ mrc1Δ*. Altogether, these data suggest that cohesin loading mediated by Chl1 is not required for the recovery from replication fork stalling.

### Chl1-dependent RPA-loading controls checkpoint activation

The fact that Chl1-dependent cohesin loading on chromatin occurs after passage of the replisome and is not involved in fork recovery suggests that Chl1 plays another critical role at stalled replication forks. It was recently proposed that Chl1 is involved in the nucleolytic resection of stalled forks (Delamarre et al, 2020). We therefore tested whether Chl1 helicase function affects RPA binding to resected ssDNA at stalled replication forks (Fig 3A). In WT, RPA accumulated at ARS607 and sites 1 and 2 kb away at 20 and 40 min after release from G1 into HU, whereas the signal decreased almost to basal levels at 60 min. Interestingly, RPA recruitment was decreased by twofold in *chl1Δ* at 20 and 40 min in HU. This decrease in RPA binding was due to loss of the helicase activity of Chl1 as RPA binding could not be rescued in *chl1Δ* expressing helicase-dead *chl1^{K48R}* (Fig 3B), consistent with a previous study (Delamarre et al, 2020). To ascertain that this was not due to impaired recruitment of Chl1^{K48R} to stalled replication forks, we complemented *chl1Δ* cells with Chl1-FLAG or Chl1^{K48R}-FLAG and examined their recruitment to HU-stalled forks. Both proteins were expressed at similar levels and bound equally well near ARS305 (Fig S1M and N), which is in accordance with previous results for ARS605, ARS606, ARS607, and ARS608 (Samora et al, 2016). Thus, the impaired RPA binding at stalled forks in *chl1Δ* was the consequence of a lack of Chl1 helicase activity. Taken together, these results suggest that Chl1's helicase activity is involved in the accumulation of ssDNA and the proper loading of RPA at stalled replication forks.

RPA-coated ssDNA is necessary to recruit the sensor kinase Mec1 to stalled replication forks in a manner dependent on its binding partner Ddc2, which is critical for Rad53-dependent checkpoint activation (Rouse & Jackson, 2002; Zou & Elledge, 2003). Therefore, we tested whether Ddc2 recruitment and Rad53 activation may be regulated by Chl1. Ddc2 efficiently accumulated at ARS607 in WT at 40 and 60 min after release from G1 into HU, whereas the signal dropped by twofold in *chl1Δ* (Fig 3C). Moreover, although the kinetics of Rad53 phosphorylation were similar in WT and *chl1Δ*, the overall levels of Rad53 phosphorylation were reduced in *chl1Δ* (Fig

3D), indicating that the formation of RPA-coated ssDNA by the Chl1 helicase affects checkpoint activation.

### Chl1 controls replication fork progression under stress conditions

Proper checkpoint activation slows down replication fork progression upon replication stress in both yeast and human cells (Seiler et al, 2007; De Piccoli et al, 2012; Somyajit et al, 2017; Bacal et al, 2018). Thus, we reasoned that impaired RPA accumulation and Rad53 phosphorylation upon *CHL1* loss may impact fork progression. To assess this, we first conducted a replication profiling analysis that allows monitoring DNA duplication in time. Both the origins and their 1 kb proximal regions were fully duplicated in WT at 40 min after release from G1 into HU (Figs 4A and S2A), whereas duplication of the same regions already occurred within 20 min in *chl1Δ* (Figs 4B and S2B). However, no difference was observed 90 min after release in HU, which suggests an increase in the rate of DNA synthesis in *CHL1*-deficient cells early after release in S phase in the presence of HU. This is in accordance with a previous study that showed that DNA replication was faster at early time points in S phase in *chl1Δ* compared with WT (Samora et al, 2016).

To corroborate this finding, we addressed whether faster DNA synthesis is also associated with faster progression of several components of the replication fork machinery (Figs 4C–E and S3A–G). In WT cells, the replicative helicase Mcm4, the leading-strand DNA polymerase Pol ε and the lagging strand primase Pol α efficiently associated with ARS607 at 20 min after release into HU, and slowly migrated over the chromosome arm within 60–90 min. Loss of *CHL1* reduced Mcm4, Pol ε and Pol α levels at sites adjacent to the origin (ARS607, +1, and +2 kb), whereas their enrichment was higher at the more distant sites (+4, +6, +7, and +8 kb) at 20 min after release into HU. Moreover, in the absence of *CHL1*, Mcm4, Pol ε, and Pol α even moved towards the 14 kb distal region at the later time points. These results indicate that in HU-treated *CHL1*-deficient cells the faster DNA synthesis is accompanied by a faster progression of the replication machinery. To assess whether the faster fork progression is due to a lack of Chl1's helicase activity, we monitored the progression of the MCM helicase at ARS607 in *chl1Δ* expressing EV, WT *CHL1* or *chl1^{K48R}* (Fig 4F). Ectopic expression of *CHL1* in *chl1Δ* rescued Mcm4 progression to normal kinetics. However, Mcm4 progression was impaired in *chl1Δ* cells expressing either *chl1^{K48R}* or the EV. These results suggest that the helicase activity of Chl1 controls replication fork rate by regulating proper formation of RPA-coated ssDNA once cells progress through S phase, thereby activating the Rad53-dependent intra-S-phase checkpoint.

To verify that the effect of Chl1 loss on replication fork progression is not due to earlier S-phase onset, we also measured recruitment of Mcm4 and Pol ε in WT and *chl1Δ* in G1 phase and at 5, 10, 15, and 20 min after release into HU (Figs S4A–E and S5A–E). Mcm4 was recruited to similar levels at ARS607 in G1 and slightly progressed after 5 and 10 min in HU in both strains (Fig S4A–C). However, fork progression was clearly faster in the absence of Chl1 after 15 and 20 min in HU with a higher enrichment of Mcm4 at the more distal loci (Fig S4D and E). This suggests that origin firing is comparable in WT and *chl1Δ*, but once origins have fired, replication forks progress faster in *chl1Δ* when compared with that in WT. Indeed, whereas Pol ε was not enriched at ARS607 in any of the two

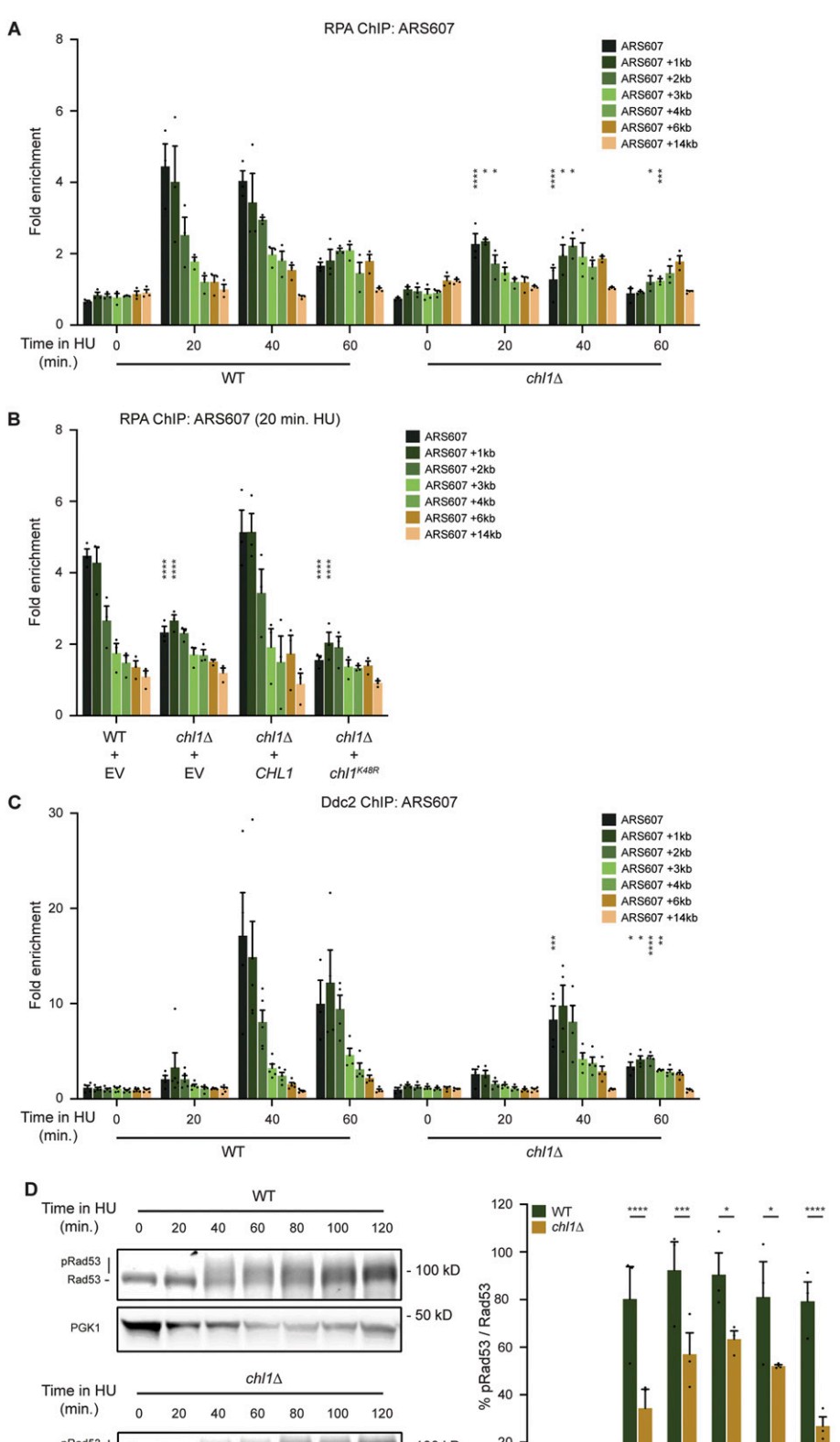

**Figure 3. Chl1 helicase controls the formation of RPA-coated single-stranded DNA and checkpoint activation.**

**(A)** Quantification of RPA recruitment by ChIP-qPCR in WT and *chl1Δ* as in Fig 2B. **(B)** As in (A), except in WT and *chl1Δ* cells expressing EV (pRS305), *CHL1* (pRS305-*CHL1*) or *chl1^K48R* (pRS305-*chl1K48R*) that were exposed to HU for 20 min. **(C)** As in (A), except for quantification of Ddc2-HA recruitment. **(D)** Western blot analysis of Rad53-FLAG phosphorylation in WT and *chl1Δ* (left). Cells were synchronized in G1 and released into YPAD containing 0.2 M HU for the indicated times. The lower band represents non-phosphorylated Rad53, whereas the upper bands represent (hyper) phosphorylated Rad53. Quantification of phosphorylated Rad53 over non-phosphorylated Rad53 (right). Error bars represent the SD of three independent experiments.

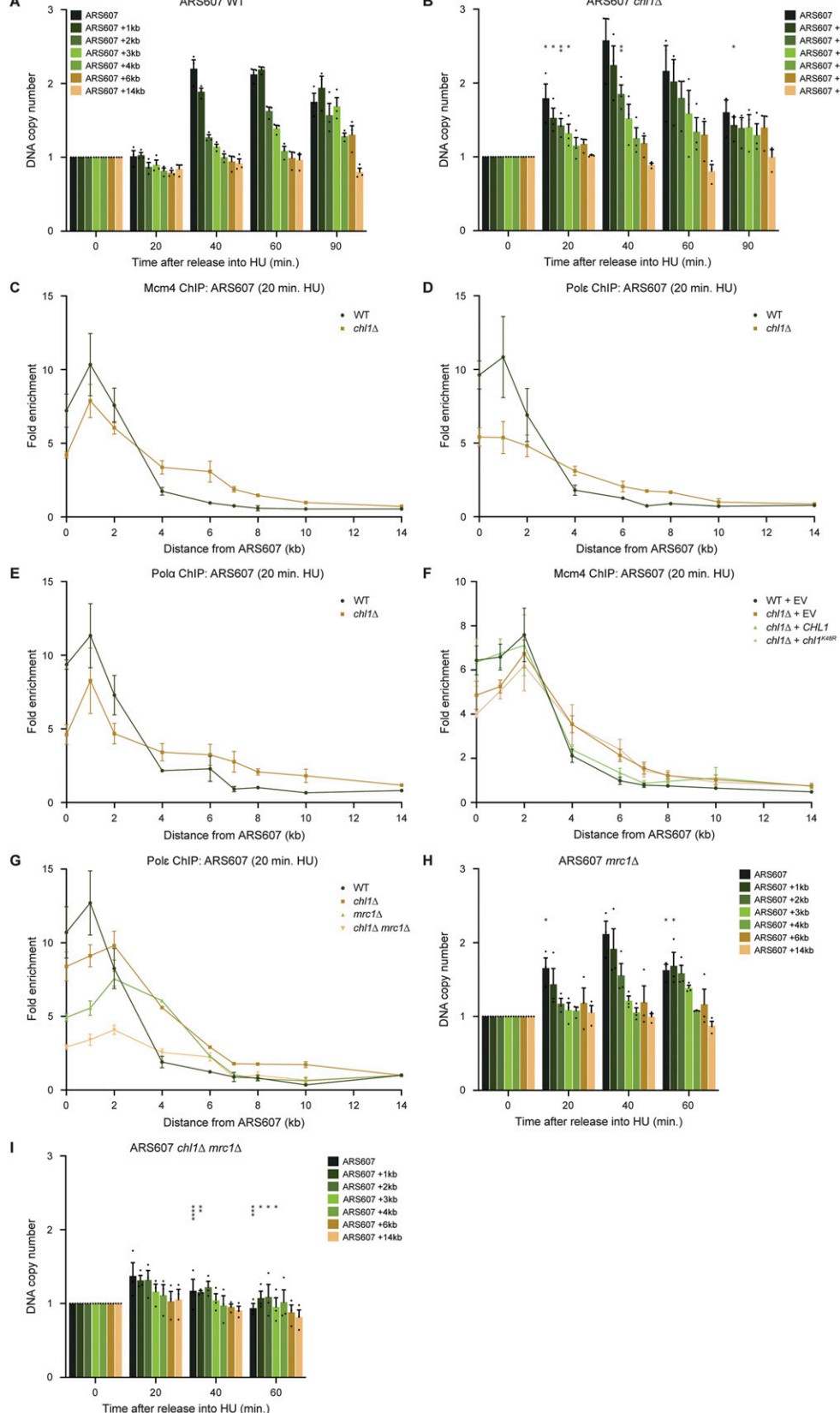

**Figure 4. Chl1 controls DNA replication speed.**
**(A, B)** Dynamics of ARS607 duplication assayed by DNA copy number analysis using qPCR in WT (A) and chl1Δ (B). Cells were synchronized in G1 phase and released into 0.2 M HU. DNA quantities were measured at the indicated time points and first normalized to that of a nonreplicated region at 60 min. The obtained ratios were then further normalized to the ratio of the samples in G1, which were set to 1. Error bars represent the standard error of the mean of three independent experiments. **(C, D, E)** Quantification of recruitment of Mcm4-FLAG (C), Pol ε-Myc (D), (B) and Pol α-HA (E) by ChIP-qPCR in WT and chl1Δ as in Fig 2B, except that cells were exposed to HU for 20 min. Error bars represent the standard error of the mean of three independent experiments. **(F)** As in (C), except in WT and chl1Δ cells expressing EV (pRS305), CHL1 (pRS305-CHL1) or chl1^{K48R} (pRS305-chl1K48R). **(G)** As in (C), except in WT, chl1Δ, mrc1Δ and chl1Δ mrc1Δ. **(H, I)** As in (A, B), except in mrc1Δ (H) and chl1Δ mrc1Δ (I). Asterisks indicate statistical differences using a t test (*P < 0.05, **P < 0.01, ***P < 0.005, ****P < 0.001).

strains, neither in G1 nor at 5 and 10 min after their release into HU (Fig S5A–C), it progressed faster in *chl1Δ* compared with WT after 15 and 20 min in HU (Fig S5D and E), which was consistent with the faster progression of Mcm4 in *chl1Δ*. We also monitored the release of WT and *chl1Δ* cells from G1 into S phase in the presence of HU by flow cytometry (Fig S5F). We did not detect any difference in S-phase entry between the two strains after 10 min in HU, strongly suggesting there is no variability in the release from G1. However, progression through S phase from 20 min on was faster in *chl1Δ* cells when compared with WT, which was consistent with faster DNA synthesis and replisome progression as determined by replication profiling and ChIP (Figs 4A–E, S2A and B, and S3A–G). Interestingly, the DNA content of *chl1Δ* cells exceeded 2C after 60 min in HU, suggesting that some endoreplication may have occurred in these cells. In conclusion, because the release from G1 and origin firing are normal upon *CHL1* loss, the increased replication fork progression in *chl1Δ* is likely not due to earlier the S-phase onset.

Given the synthetic relationship between *CHL1* and *MRC1*, we also assessed the progression of Pol ε at ARS607 in *mrc1Δ* and *chl1Δ mrc1Δ* 20 min after release from G1 into HU (Fig 4G). Pol ε moved further away from ARS607 in *mrc1Δ* when compared with WT, resembling the phenotype of *chl1Δ*. However, whereas faster movement of Pol ε was accompanied by enhanced DNA synthesis rates in *chl1Δ*, only a slightly reduced DNA synthesis rate was observed in *mrc1Δ* at both ARS607 and ARS305 (Figs 4H and S2C), in agreement with previous studies (Katou et al, 2003; Lou et al, 2008). Strikingly, in *chl1Δ mrc1Δ* Pol ε was nearly absent from ARS607 (Fig 4G). In addition, DNA synthesis was barely detectable at ARS607 and ARS305, not even at 60 min after release into HU (Figs 4I and S2D),

suggesting that replication progression is defective in *chl1Δ mrc1Δ*. Altogether, this argues that Chl1 and Mrc1 act synergistically to control replication fork progression under replication stress conditions.

## Chl1 loss affects dNTP levels without activating the DNA damage response

We wondered whether the increase in fork progression upon *CHL1* loss was only a consequence of the defect in nucleolytic processing of stalled forks, or also a result of increased dNTP pools, a phenomenon known to enhance DNA synthesis in the presence of HU (Poli et al, 2012). To examine if *CHL1* loss affects fork progression by impacting dNTP pools, we determined the levels of dGTP, dATP, dCTP, and dTTP in WT, *chl1Δ*, *mrc1Δ*, and *chl1Δ mrc1Δ* cells in G1 and 60 min after release into HU (Fig 5A). *sml1Δ* cells, which served as a positive control, showed a strong increase in the dNTP levels (Zhao et al, 1998). The dNTP levels were higher in *chl1Δ* compared with the WT, but similar to those in *mrc1Δ*, which did not show faster DNA synthesis. Moreover, we observed a strong increase in dNTP levels in *chl1Δ mrc1Δ*, although DNA synthesis was almost abrogated in this strain.

To determine whether the increase in dNTP levels upon loss of Chl1 is dependent on its helicase activity, we measured dNTP levels in *chl1Δ* complemented with EV, WT *CHL1*, or *chl1^{K48R}* (Fig 5B). Whereas re-expression of WT *CHL1* lowered the dNTP levels in G1 phase to those observed in the WT strain, expression of *chl1^{K48R}* did not. Thus, Chl1 controls dNTP levels in a manner dependent on its helicase activity.

Because DNA damage leads to increased dNTP pools, we wondered if the elevated dNTP levels in G1 phase observed after chronic *CHL1* loss may be a consequence of spontaneous DNA damage that

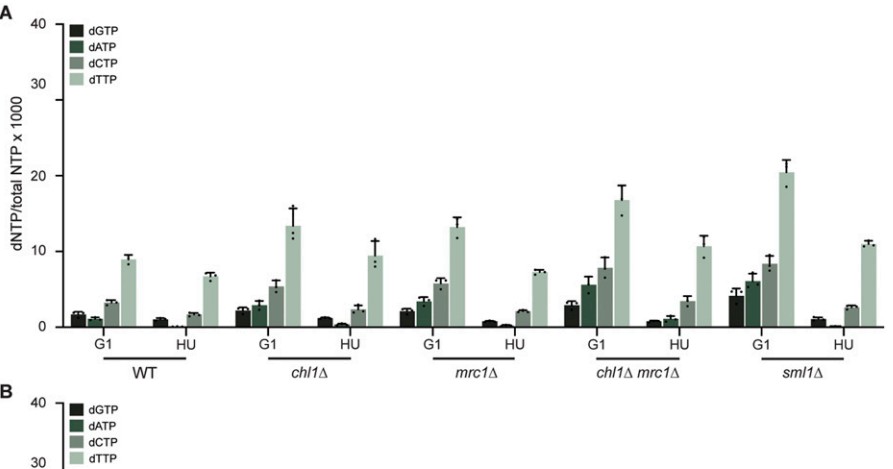

**Figure 5. Chl1 helicase regulates dNTP levels.**
**(A)** Analysis of dNTP concentrations in WT, *chl1Δ*, *mrc1Δ*, *chl1Δ mrc1Δ* and *sml1Δ* synchronized in G1 phase or 60 min after release from G1 into 0.2 M HU. Error bars represents the SD of three independent experiments. **(B)** As in (A), except in WT and *chl1Δ* expressing EV (pRS305), *CHL1* (pRS305-*CHL1*), or *chl1^{K48R}* (pRS305-*chl1K48R*).

accumulated during previous cycles of DNA replication (Crabbe et al, 2010; Davidson et al, 2012; Poli et al, 2012). To this end, we examined the formation of Rad52 foci, which form at low frequency and mark damaged replication forks in unperturbed S-phase cells (Lisby et al, 2001, 2003). However, Rad52 foci levels were similar in WT and chl1Δ (Fig S6A and B). In addition, the expression of HUG1, whose transcription is dependent on replication arrest and DNA damage (Basrai et al, 1999), was neither induced in WT nor in chl1Δ G1-phase cells, whereas it was similarly induced after 60 min in HU in both strains (Fig S6C). Moreover, Rad53 was not activated in untreated chl1Δ G1-phase cells (Fig 3D; t0), refuting any chronic checkpoint activation in this strain. Therefore, loss of CHL1 helicase function impacts dNTP pools likely without activating the DNA damage response.

## Chl1 suppresses RNR1 expression to prevent accelerated fork progression

To assess whether Chl1 impacts dNTP levels and RPA recruitment via distinct pathways that both contribute to replication fork progression, we first assessed whether increased dNTPs levels affect RPA binding near HU-stalled forks in WT, chl1Δ and sml1Δ mutants (Fig S7A). Strikingly, we observed impairment RPA binding in chl1Δ, which was even more pronounced in chl1Δ, suggesting that the accumulation of dNTPs results in reduced formation of RPA-coated ssDNA. Next, we measured dNTP levels in WT, chl1Δ, sml1Δ and sml1Δ chl1Δ cells in G1 and 60 min after release into HU (Fig 6A). Interestingly, we noticed an additive effect on the increase in dNTP levels in sml1Δ chl1Δ when compared with that in each single mutant, suggesting independent modes of dNTP regulation by Sml1 and Chl1.

We then measured DNA synthesis at ARS607 by replication profiling analysis in these strains. In accordance with a previous study (Poli et al, 2012), DNA replication was faster in sml1Δ, as loci at 2 and 3 kb away from ARS607 were duplicated within 40 min in this strain, whereas these same regions were only replicated after 60 min in WT (Fig 6B and C). In sml1Δ chl1Δ, DNA replication was faster than in sml1Δ or chl1Δ (Fig 6C–E), suggesting that fork progression is further enhanced by the increase in dNTP levels because of CHL1 loss. Therefore, we concluded that Chl1 regulates fork progression by controlling proper dNTP levels.

dNTP levels are regulated by the ribonucleotide reductase (RNR) complex, whose subunits are encoded by four genes, RNR1, RNR2, RNR3 and RNR4. Transcription of these genes is tightly controlled in G1 phase, before their up-regulation in S phase in response to DNA damage (Chabes & Stillman, 2007). Given the increase in dNTP levels after loss of CHL1, we examined whether CHL1 controls the expression of RNR genes before S-phase entry (Figs 6F and S7B–D). Interestingly, expression of RNR1, but not RNR2, RNR3 or RNR4 was up-regulated in chl1Δ expressing EV or chl1^{K48R} when compared with that in WT or chl1Δ expressing CHL1, indicating that Chl1 regulates RNR1 transcription through its helicase activity. To rule out cell cycle effects due to chronic loss of CHL1, we also measured the expression of the four RNR genes after transient depletion of Chl1 using a strain expressing Auxin-Inducible Degron (AID)-tagged Chl1. Importantly, we observed efficient Chl1 degradation after 5 h of auxin treatment (Fig S7E), concomitantly with an increase in RNR1 expression (Fig S7F), ruling out indirect effects from previous cell cycles.

To further explore the link between Chl1 and RNR1, we next asked whether Chl1 might also regulate the expression of SML1 (Fig S7G),

which strongly inhibits Rnr1 by direct binding (Zhao et al, 1998). However, we could not detect any difference in SML1 transcript levels in chl1Δ compared to WT, consistent with the additive effect of the sml1Δ chl1Δ double mutant on dNTP levels (Fig 6A). Finally, we assessed if Chl1 might bind to the RNR1, RNR2, RNR3 or RNR4 promoter in G1-phase cells. However, we could not detect Chl1 binding at any of the RNR promoters, although Chl1 showed binding near ARS607 in HU-treated cells as expected, showing the validity of the approach (Fig S7H). Altogether, these data indicate a novel pathway for RNR1 regulation through the helicase function of Chl1, which controls dNTP levels likely independently of Sml1 regulation or the accumulation of spontaneous DNA damage.

# Discussion

Systematic mapping of synthetic genetic interactions in yeast revealed an interaction between CHL1 and MRC1 (Sun et al, 2020), a known gene involved in the replication stress response, suggesting a role for CHL1 in this response. Here, we reveal that Chl1, specifically through its helicase activity, controls the replication stress response by regulating the dNTP levels through RNR1 gene expression at the onset of S phase. This controls replication fork progression after entry of cells into S phase by favouring the formation of RPA-coated ssDNA at stalled replication forks, and the subsequent activation of a Rad53-dependent checkpoint. Proper loading of RPA has also been shown to promote cohesin recruitment to stalled replication forks (Fig 6G) (Delamarre et al, 2020).

Although the phenotypes we observed in Chl1-deficient cells seem to result from the effect of increased RNR1 gene expression and the subsequent increase in dNTP levels in G1, an important question is whether the Chl1 helicase also plays a direct role at stalled replication forks. The fact that this helicase associates with the replication machinery in both yeast and human cells and is present at replication forks under normal and replication stress conditions argues for an active role at these structures (our data and Samora et al [2016] and Cortone et al [2018]). Biochemical characterization of the human counterpart of Chl1, DDX11, showed that it progresses along ssDNA with a 5′-3′ directionality (Hirota & Lahti, 2000; Farina et al, 2008), suggesting that it may favour DNA unwinding on the lagging strand. Moreover, DDX11 can process a variety of secondary structures, including forked or 5′ flap duplexes, triplex and G-quadruplexes (Wu et al, 2012; Bharti et al, 2013, 2014; Lerner et al, 2020; van Schie et al, 2020). Accordingly, human cells depleted of DDX11 showed reduced replication fork speed, impaired cohesin loading and defective SCC (Cali et al, 2016; Faramarz et al, 2020; van Schie et al, 2020) due to their inability to process G4 structures during replication (Lerner et al, 2020; van Schie et al, 2020). Whether Chl1 is also able to process such structures and whether this underlies the slower replication speed observed in chl1Δ at late time points after HU, as well as the observed cohesin loading defects in this strain, remains to be established (Samora et al, 2016). Alternatively, both our work and that of others suggests that Chl1/DDX11 may also promote nascent DNA resection at stalled forks, generating RPA-coated ssDNA that is required to sustain checkpoint activation and facilitate cohesin loading (Delamarre et al, 2020; Simon et al, 2020). Thus, the available data seem to suggest

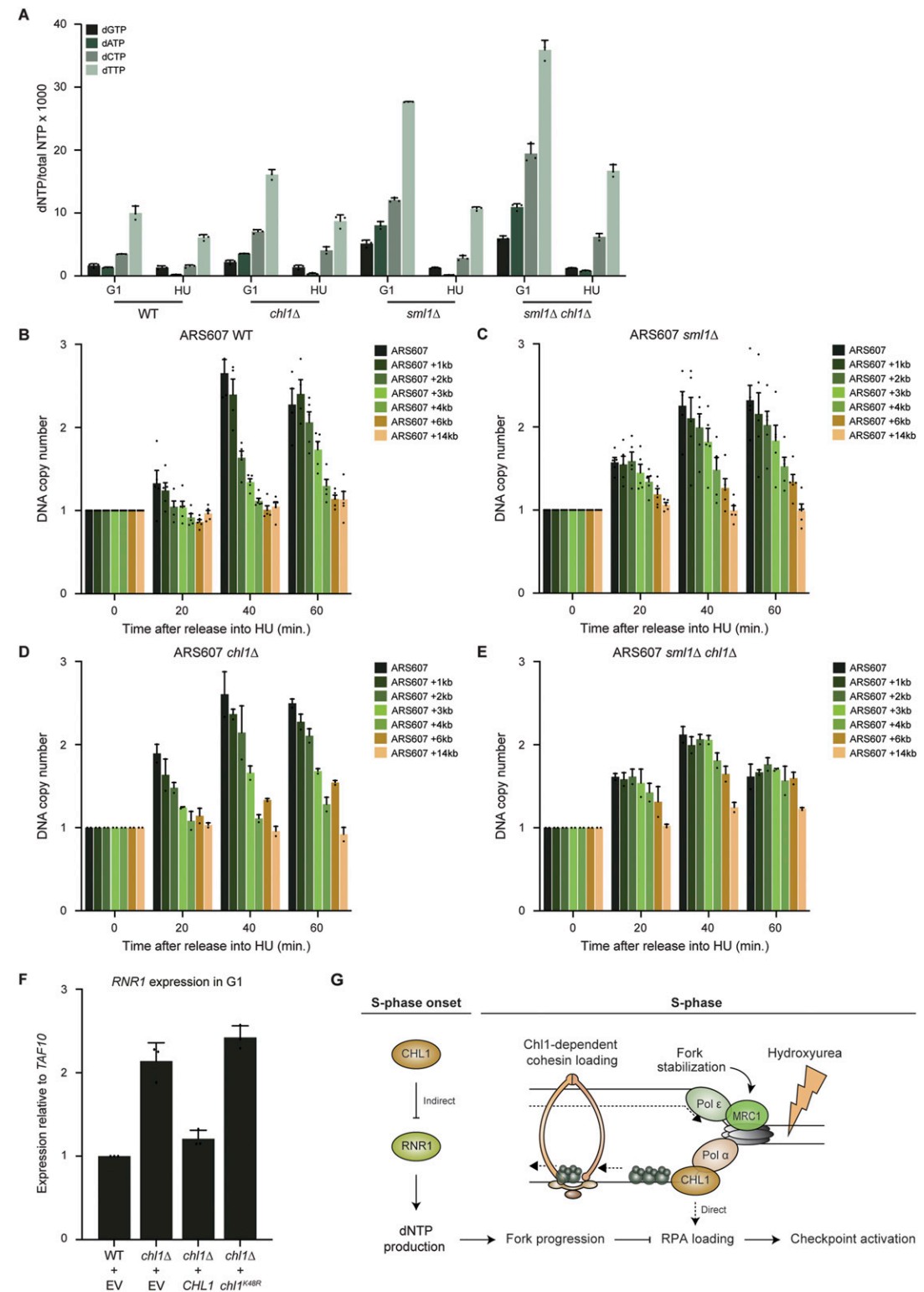

**Figure 6. Chl1 helicase controls replication fork progression by modulating *RNR1* expression and dNTP pools.**
**(A)** Analysis of dNTP concentrations as in Fig 5A, except in WT, *chl1Δ*, *sml1Δ* and *sml1Δ chl1Δ*. **(B, C, D, E)** Dynamics of ARS607 duplication assayed by DNA copy number analysis using qPCR as in Fig 4A, except in WT (B), *sml1Δ* (C), *chl1Δ* (D) and *sml1Δ chl1Δ* (E). Error bars represent the standard error of the mean of two independent experiments. **(F)** Quantification of *RNR1* gene expression by RT-qPCR in WT and *chl1Δ* expressing EV (pRS305), *CHL1* (pRS305-*CHL1*) or *chl1K48R* (pRS305-*chl1K48R*) and synchronized in G1. Error bars represents the SD of three independent experiments. **(G)** Model showing distinct roles of Chl1 and Mrc1 at stalled replication forks. Chl1 and Mrc1 are both recruited to stalled replication forks. Mrc1 stabilizes the replisome at arrested replication forks and participates in checkpoint activation. Chl1's helicase activity regulates *RNR1* expression and dNTP pools at the onset of S phase. This controls replication fork progression by affecting the formation of RPA-coated

that Chl1 impacts fork stability and progression in response to HU directly by acting at stalled forks, and indirectly by controlling dNTP levels at the onset of S phase (Fig 6G).

Next, we questioned to what extent these findings can explain the synergistic effect of Chl1 and Mrc1 loss during the replication stress response. Whereas Mrc1 is required to ensure the coupling between Pol ε and the replicative helicase Mcm2-7 (Katou et al, 2003; Gambus et al, 2006; Lou et al, 2008; Bando et al, 2009; Komata et al, 2009), thereby stabilizing the replication machinery to promote DNA replication (Katou et al, 2003; Szyjka et al, 2005; Tourriere et al, 2005), Chl1 rather controls replication fork speed. Moreover, we found that Chl1 and Mrc1 differentially activate the intra-S checkpoint. Whereas loss of Mrc1 delays checkpoint activation (Alcasabas et al, 2001; Osborn & Elledge, 2003), Chl1 does not affect the kinetics, but rather promotes activation of the checkpoint. Finally, although it was proposed that Chl1 and Mrc1 are involved in two complementary pathways of cohesion establishment (Xu et al, 2007), we demonstrated that Mrc1, contrary to Chl1, is dispensable for cohesin loading at stalled forks. These findings illustrate the opposing roles that Chl1 and Mrc1 play in controlling DNA replication and checkpoint activation after replication stress, explaining their synergistic impact on the replication stress response. However, further research will be needed to investigate whether in the case of Chl1 this synergistic effect relies on its direct and/or indirect role in response to fork stalling.

dNTP pools are regulated by modulating the activity of ribonucleotide reductases by multiple mechanisms. Under normal conditions, three out of the four *RNR* genes, *RNR2*, *RNR3* and *RNR4*, are transcriptionally repressed by Ctr1 (Huang & Elledge, 1997; Huang et al, 1998). The fourth one, *RNR1*, is bound by the Sml1 inhibitor protein (Zhao et al, 1998). All *RNR* genes are induced by DNA damage, which leads to an increase in dNTP levels and fork progression most notably in mutants with chromosomal instability (CIN) (Crabbe et al, 2010; Davidson et al, 2012; Poli et al, 2012). Here, we uncover a novel mechanism by which Chl1 controls the dNTP levels, which is likely independent of Sml1 or the accumulation of spontaneous DNA damage/CIN. Chl1 rather controls the expression of *RNR1* in G1 phase through its helicase activity. The exact mechanism by which the Chl1 helicase inhibits *RNR1* transcription remains elusive, but warrants further investigation given its impact on replication fork stability/progression.

# Materials and Methods

## Yeast growth conditions, strains, and plasmids

All yeast strains are derivatives of W303-1A (Table S1). Cells were grown in rich YPA medium supplemented with 2% glucose (YPAD) or 2% raffinose (YPARaff). To induce gene expression from the *GAL1* promoter, cells were grown in raffinose-containing medium before addition of 2% galactose. Gene deletions and epitope tags on endogenous genes were generated by PCR-based gene targeting (Longtine et al, 1998). To acutely deplete Chl1, *CHL1* was AID-tagged in a strain in which the OsTIR1 expression cassette was integrated at

the *TRP1* locus. Chl1 was depleted by adding 2 mM 3-IAA (Sigma-Aldrich) for 5 h (Nishimura et al, 2009). Primers are available upon request. The *CHL1* ORF was amplified from genomic DNA with primers containing *SalI* and *SacII* restriction sites and the digested PCR product was ligated into the pRS305 *SalI* and *SacII* sites to obtain pRS305-*CHL1*. pRS305-*chl1*$^{K48R}$ was generated by introducing the K48R mutation into pRS305-*CHL1* by site-directed mutagenesis using the QuikChange Lightning Site-Directed Mutagenesis Kit (Agilent) with primers changing the respective AGA codon to GAA. Plasmids pRS305, pRS305-*CHL1* and pRS305-*chl1*$^{K48R}$ were digested with *AflII* to introduce them at the *LEU2* locus in the yeast genome. *CHL1* and *chl1*$^{K48R}$ were excised from pRS305-*CHL1* and pRS305-*chl1*$^{K48R}$ by *SalI* and *SacII* digestion, and ligated into the pRS303 *SalI* and *SacII* sites to generate pRS303-*CHL1* and pRS303-*chl1*$^{K48R}$, respectively. Plasmids pRS303, pRS303-*CHL1* and pRS303-*chl1*$^{K48R}$ were digested with *NheI* to introduce them at the *HIS3* locus.

## Spot dilution test

Cells were grown overnight in YPAD and then plated in 10-fold serial dilutions starting at a density of $1.2 \times 10^7$ cells/ml (OD$_{600 \text{ nm}}$ = 1) or in fivefold serial dilutions starting at a density of $0.6 \times 10^7$ cells/ml (OD$_{600 \text{ nm}}$ = 0.5) on YPAD plates without or with 50 or 100 mM HU. Cells were grown for 3 d at 30°C before images were taken.

## HU recovery assay

Cells were grown to $1 \times 10^6$ cells/ml in YPA containing 2% raffinose (YPARaff) and released in rich YPA medium containing 0.2 M HU and supplemented with 2% glucose (YPAD) or 2% galactose (YPAGal) for 8 h. Each culture was appropriately diluted and plated on YPAD or YPAGal. Colonies were counted after 3 d of incubation at 30°C.

## Chromatin immunoprecipitation

ChIP was performed as described (Cobb et al, 2003). Cells were grown to $5 \times 10^6$ cells/ml in YPAD, synchronized with α-factor for 2 h, washed in YPAD and released in YPAD containing 0.2 M HU (Sigma-Aldrich). Samples were collected at 0, 20, 40, 60, and 90 min after release and fixed with 1% formaldehyde. Alternatively, cells were released in YPAD at 20°C, and samples were collected at 30 and 40 min after release and fixed with 1% formaldehyde. Extracts were subjected to immunoprecipitation using Dynabeads coated with mouse or rabbit IgG (M-280; Invitrogen) in combination with antibody against c-Myc (9B11; Santa Cruz), Flag (clone M2; Sigma-Aldrich), HA (clone HA-7; Sigma-Aldrich) or RPA (AS07 214; Agrisera). DNA was purified and enrichment at specific loci was measured using qPCR. Primers used are listed in Table S2. Relative enrichment was determined by the $2^{-\Delta\Delta Ct}$ method (Livak & Schmittgen, 2001; Cobb & van Attikum, 2010). Signal for Dynabeads alone was used to correct for background. An amplicon at the *SMC2* locus or 14 kb downstream of ARS607 was used as endogenous control for ChI the ChIP of replication protein. An amplicon 11 kb downstream of ARS305, devoid from

---

single-stranded DNA and the subsequent activation of Rad53 checkpoint kinase. Moreover, the formation of RPA-coated single-stranded DNA by Chl1 also favours cohesin loading at stalled forks. Asterisks indicate statistical differences using a *t* test (*P < 0.05, **P < 0.01, ***P < 0.005, ****P < 0.001).

Scc1 binding was used for normalization in ChIP cohesin (Tittel-Elmer et al, 2012).

### Replication timing analysis

Cells were grown to $5 \times 10^6$ cells/ml in YPAD, synchronized with $\alpha$-factor for 2 h, washed in YPAD and released in YPAD containing 0.2 M HU (Sigma-Aldrich). $2.5 \times 10^7$ cells were collected at 0, 20, 40, 60, and 90 min after release from G1, fixed with 0.2% Na-Azide for 10 min and washed with 10 mM Tris, 50 mM EDTA. For genomic DNA extraction, cells were digested in 1 M sorbitol, 0.1 M sodium citrate, pH 7.0, 60 mM EDTA, 8 mg/ml $\beta$-mercaptoethanol, 2 mg/ml zymolyase 20T for 45 min, and DNA was isolated using the QIAGEN DNeasy Blood and Tissue kit following the manufacturer's instructions. The amount of genomic DNA at ARS607 and downstream loci was quantified by qPCR using the ratio of DNA in HU-arrested to that in G1 and normalized to *TAF10* locus.

### dNTP quantification

dNTP quantification was performed as previously described (Watt et al, 2016).

### Western blot

Rad53-FLAG expressing cells were grown to $5 \times 10^6$ cells/ml in YPAD, synchronized with $\alpha$-factor for 2 h, washed with YPAD and released in YPAD containing 0.2 M HU (Sigma-Aldrich). Samples were collected at 0, 20, 40, 60, 80, 100 and 120 min after release. Chl1-FLAG and Chl1$^{K48R}$-FLAG expressing cells were grown to $5 \times 10^6$ cells/ml in YPAD. Chl1-Myc-AID expressing cells were grown to $5 \times 10^6$ cells/ml in YPAD and synchronized with $\alpha$-factor for 2 h. Whole cell extracts were prepared by TCA precipitation and analyzed by SDS–PAGE. Western blotting was performed using an anti-Flag antibody (clone M2; Sigma-Aldrich) or anti-Myc antibody (Santa Cruz [9E10: sc-40]). An anti-PGK1 antibody (22C5D8; Invitrogen) was used to control protein loading.

### Co-immunoprecipitation

Samples were prepared as previously described (Maric et al, 2014) with the following modifications. Briefly, after resuspension in lysis buffer (100 mM HEPES-KOH, pH 7.9, 50 mM potassium acetate, 10 mM magnesium acetate, and 2 mM EDTA), frozen yeast cells were ground manually. Insoluble cell debris were pelleted at 21,000*g* for 30 min before samples were incubated with Dynabeads coated with mouse IgG (M-280; Invitrogen) and antibody against Flag (clone M2; Sigma-Aldrich), c-Myc (9B11; Santa Cruz), HA (clone HA-7; Sigma-Aldrich), RPA (AS07 214; Agrisera) or mouse IgG (02-6502; Invitrogen).

### RT-qPCR

Cells were grown to $5 \times 10^6$ cells/ml in YPAD and synchronized with $\alpha$-factor for 2 h. $1.5 \times 10^7$ cells were harvested and RNA was isolated with the RNeasy Mini kit according to the manufacturer's protocol (QIAGEN). Genomic DNA was digested with TURBO DNA-free Kit (Invitrogen) or with the RNase Free DNase set (QIAGEN) and subsequently RNA was purified with the RNeasy Mini kit (QIAGEN). cDNA preparation was performed using the GoScript Reverse Transcriptase System (Promega). The expression levels of the *RNR*, *HUG1* and *SML1* genes were quantified by qPCR and normalized to *TAF10* locus. Primers used are listed in Table S2.

### Rad52 foci analysis

Cells were grown to mid-log in YPAD and fixed in 4% paraformaldehyde at room temperature for 15 min, washed, and resuspended in $KPO_4$/Sorbitol solution (10 mM $KPO_4$, 1.2 M Sorbitol, pH = 7.5). Images were acquired on a Zeiss AxioImager M2 widefield fluorescence microscope equipped with 100x PLAN APO (1.4 NA) oil-immersion objectives (Zeiss) and an HXP 120 metal-halide lamp used for excitation. 21 focal steps of 0.25 $\mu$m were acquired with an exposure time of 1,000 ms using a GFP/YFP 488 filter (excitation filter: 470/40 nm, dichroic mirror: 495 nm, emission filter: 525/50 nm). Images were recorded using ZEN 2012 software and analyzed with Fiji.

## Supplementary Information

## Acknowledgements

We thank Susan Gasser, Jennifer Cobb, Maria-Pia Longhese, Lorraine Symington, Karine Dubrana, Philippe Pasero, Richard Kolodner, and Rodney Rothstein for providing yeast strains and plasmids. We thank Philippe Pasero, Armelle Lengronne, Rob Wolthuis, and Job de Lange for insightful discussions. This work was financially supported by grants from the Netherlands Organization for Scientific Research (NWO TOP-GO – 85410013 and NWO VICI – 182.052) and the European Research Council (ERC Consolidator grant – 617485) to H van Attikum, and the Swedish Cancer Society and the Swedish Research Council to A Chabes.

### Author Contributions

A Batté: conceptualization, data curation, formal analysis, validation, investigation, visualization, methodology, and writing—original draft, review, and editing.
SC van der Horst: data curation, formal analysis, validation, investigation, methodology, and writing—review and editing.
M Tittel-Elmer: conceptualization, data curation, formal analysis, validation, investigation, methodology, and writing—review and editing.
SM Sun: conceptualization, data curation, formal analysis, validation, investigation, and methodology.
S Sharma: data curation and investigation.
J van Leeuwen: resources.
A Chabes: resources and funding acquisition.
H van Attikum: conceptualization, resources, supervision, funding acquisition, visualization, methodology, project administration, and writing—review and editing.

## Conflict of Interest Statement

The authors declare that they have no conflict of interest.

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
