## [Reviewer comments · Life Science Alliance]

Life Science Alliance

Chl1 helicase controls replication fork progression by regulating dNTP pools

Amandine Batte, Sophie van der Horst, Mireille Tittel-Elmer, Su Ming Sun, Sushma Sharma, Jolanda Van Leeuwen, Andrei Chabes, and Haico van Attikum

DOI: <https://doi.org/10.26508/lsa.202101153>

Corresponding author(s): Haico van Attikum, Leiden University Medical Center and Amandine Batte, University of Lausanne

Review Timeline:

Submission Date:	2021-07-08
Editorial Decision:	2021-08-23
Revision Received:	2021-11-24
Editorial Decision:	2021-12-20
Revision Received:	2021-12-23
Accepted:	2021-12-28

Scientific Editor: Novella Guidi

Transaction Report:

August 23, 2021

Re: Life Science Alliance manuscript #LSA-2021-01153-T

Prof. Haico van Attikum
Leiden University Medical Center
Department of Human Genetics
Einthovenweg 20
Leiden 2333 ZC
Netherlands

Dear Dr. van Attikum,

Thank you for submitting your manuscript entitled "Chl1 helicase controls replication fork progression by regulating dNTP pools and checkpoint activation" to Life Science Alliance. The manuscript was assessed by expert reviewers, whose comments are appended to this letter. All referees are quite positive and excited about this work that in their views it is carried out to a very high standard and provide additional validation of current models. Besides Reviewer 2 and 3 which point out to changes to address only in the text and figures, Reviewer 1 main concern is that the data supporting a transcriptional control function for Chl1 on of RNR1/dNTP pools is not sufficiently supported by the data currently presented and there is the risk of misinterpretation, therefore it would be key to provide additional support for the proposed model. Thus, additional evidence for a transcription regulation role of Chl1 (and clear distinction from checkpoint-mediated dNTP up-regulation as an alternative explanation for the observations presented) would be required. This reviewer suggests a couple of experiments to conduct that will strengthen the Chl1-RNR1 link proposed in the manuscript. In case you won't be able to perform these experiments please address these issues further in the text before resubmitting a revised version of the manuscript. All the other concerns raised by the reviewers should be addressed as well. We, thus, encourage you to submit a revised version of the manuscript back to LSA that responds to all of the reviewers' points.

Thank you for this interesting contribution to Life Science Alliance. We are looking forward to receiving your revised manuscript.

Sincerely,

B. MANUSCRIPT ORGANIZATION AND FORMATTING:

Reviewer #1 (Comments to the Authors (Required)):

Betté and co-workers address the role of the Chl1 (DDX11) helicase in DNA replication fork progression under replication stress conditions in budding yeast. This has recently been studied by Delamarre et al., *Mol Cell* 77, 395-410 (2020), who have shown that the Chl1 helicase activity is required for cohesin loading and nascent DNA resection at stalled DNA replication forks. Thus, Chl1/DDX11 activity contributes to ssDNA formation and RPA association/checkpoint signalling at stalled forks, and ultimately fork restart (Delamarre et al., 2020; Simon et al., *Life Sci Alliance* 3, e201900547 (2020)). This mechanism is distinct from the canonical role of Chl1 in sister chromatid cohesion establishment independently of its helicase activity during unperturbed DNA replication (Samora et al., *Mol Cell* 63, P371-384 (2016)). The results in the current MS by Betté et al. are consistent with the Samora, Delamarre, and Simon studies, and similar conclusions are drawn.

Betté and co-workers have previously detected a negative genetic interaction between chl1 and mrc1 (Ming Sun et al., *J Cell Sci* 133, jcs237628 (2020), see also Costanzo et al., *Science* 353, pii: aaf1420 (2016)), leading to the hypothesis of a contribution of Chl1 to the replication stress response. This genetic interaction is apparent as a growth defect in unperturbed conditions and synthetic hypersensitivity to dNTP depletion induced by HU. The authors show (Fig. 1) that loss of Chl1 negatively synergises with a slow replication and a checkpoint signalling mutant of Mrc1, while growth defects in unperturbed conditions and HU hypersensitivity in complete absence of Mrc1 and WT Chl1 is not rescued by expression of a helicase-dead version of Chl1. Since Chl1's canonical cohesion loading function does not depend on its helicase activity (Samora et al., *Mol Cell* 63, P371-384 (2016)), this implies that growth/replicative defects in mrc1 chl1 double-mutant cells cannot be explained by Chl1's role in cohesion loading in unperturbed conditions. This notion is elaborated further (Fig. 2) by providing evidence through cohesion ChIP experiments that, as expected, Chl1 is required for cohesion loading during replication, but that over-expression of SCC1 (previously shown to overcome the chl1 cohesion loading defect, Samora et al., *Mol Cell* 63, P371-384 (2016)), does not suppress a loss of viability observed for mrc1 chl1 double deletion mutants after a prolonged period of replication arrest. Replication profiling and replication factor ChIP analyses (Fig. 3) are used to show that replication in the presence of HU proceeds faster in chl1 mutant cells, which is consistent with DNA combing experiments performed by the Uhlmann lab (see Fig. 4C in Samora et al., *Mol Cell* 63, P371-384 (2016)) showing directly a statistically significant increase in nascent DNA track length in the presence of HU after 90 min when Chl1 is absent. This may relate in part to elevated G1 dNTP levels in chl1 mutants detected by Betté and co-workers (Fig. 4), which coincides with increased RNR1 expression levels in chl1 deletion or helicase-dead mutants. Finally, a dependence of RPA accumulation at stalled forks on the Chl1 helicase first described by Delamarre and co-workers can be confirmed (Fig. 5), and a reduction in RPA correlates with a reduction in checkpoint activation as detected by Rad53 phosphorylation, consistent with RPA and ssDNA providing the landing pad for the apical kinase Mec1-Ddc2, and also consistent with mammalian data (Simon et al., *Life Sci Alliance* 3, e201900547 (2020)). The authors conclude with a model (Fig. 5F) summing up (1) the role of Chl1 in RPA establishment at stalled DNA replication forks as previously described and (2) add a second aspect, namely a Chl1-dependent downregulation of RNR1 at S phase onset leading to reduced dNTP levels, and therefore reduced initial replication speed.

Although the current MS overlaps largely with recent reports on the role of Chl1 at stalled DNA replication forks, the work is carried out to a very high standard and provides additional validation of current models (Delamarre et al., 2020; Simon et al., *Life Sci Alliance* 3, e201900547 (2020)). There are some overstatements in the current version of the MS that should be addressed, and additional evidence is critical to support a regulatory role of the Chl1 helicase on RNR1 expression.

Major points:

-There is probably one aspect in which the current MS goes beyond the current literature on Chl1 functions. This is the potential involvement of the Chl1 helicase activity in RNR1 expression and therefore dNTP levels (see model Fig. 5F). At present the data supporting a transcriptional control function for Chl1 is not very strong and it would be key to provide additional support for the proposed model. The authors point out the caveat that dNTP levels may simply be deregulated in chl1 mutants due to elevated intrinsic replication stress levels. This is addressed by showing that Rad52 foci are not elevated in chl1 deletion mutants, but the question remains whether this assay is sensitive enough to detect elevated replication stress and the possibility that dNTP levels may be deregulated because of that. It appears that chl1 mutants have a cell morphology indicative of replication stress and damage and large dumbbell shaped cells are apparent (Fig. EV6D). Is this concomitant with chronic checkpoint activation? To rule out multi-cell cycle effects caused by a CHL1 deletion, it may be possible to use a degradation tag and acutely deplete Chl1, perhaps in G1, and monitor effects on RNR1 expression. An alternative approach could be to experimentally titrate Chl1 levels and assess how RNR1 expression responds. While this would not mechanistically resolve how Chl1 helicase activity may play a role in RNR1 gene expression, these experiments, or perhaps other supporting data the authors may be able to provide, would be helpful to strengthen the Chl1-RNR1 link proposed in the MS.

-Key points in the discussion are similar to those in previous work, in particular in the Delamarre 2020 paper mentioned above. A clear description of the state-of-the art of Chl1 at stalled replication forks, summarizing the key papers (Delamarre et al., 2020; Simon et al., Life Sci Alliance 3, e201900547 (2020); Samora et al., Mol Cell 63, P371-384 (2016)) might be very helpful for readers. This would allow to clearly state which aspects of the work are of a more confirmatory nature, and to then focus on more novel aspects, i.e., the potential link between Chl1 helicase activity and RNR1 expression and the benefits this mechanism might have.

Minor points:

-The experiment shown in Fig. 1D should be carried out with the mrc1 separation-of-function mutants used in 1C to support the statement at the bottom of p. 7 that "Chl1's helicase activity is required for checkpoint control and replication fork progression."

-Could the authors clarify how the experiment in Fig. 2F showing that prolonged exposure of cells to 200 mM HU results in a significant loss of viability in the absence of Mrc1 or Chl1/Mrc1 (there is only little effect on viability when only Chl1 is absent under the conditions employed), and that this effect is not suppressed by overexpression of Scc1, warrants the conclusion on p. 10 that "cohesin loading mediated by Chl1 is not involved in the recovery from replication fork stalling"? On face value it appears that Scc1 over-expression under these conditions is not sufficient to promote DNA replication fork stability and/or restart when Mrc1 is absent.

-The statement on p. 10 with regard to replication profiling that "no difference was observed 90 minutes after release from HU, in accordance with a previous study (Samora et al., 2016)" is slightly misleading. The Samora study clearly shows that replication in the presence of HU proceeds faster when Chl1 is absent at the 90 min time-point after release of synchronized cells into 200 mM HU (see Fig. 4C in Samora et al., Mol Cell 63, P371-384 (2016)). Having said this, the trends in the Samora study and the current MS are similar. Perhaps this can be rephrased to avoid confusion.

Reviewer #2 (Comments to the Authors (Required)):

Batte et al examine the role of the DNA helicase Chl1 in the DNA replication stress response. Chl1 has previously been connected to sister chromatid cohesion, in a pathway parallel to the checkpoint mediator Mrc1. The authors start from the observation that chl1Δ and mrc1Δ have a strong negative genetic interaction, following up with separation of function alleles to suggest that CHL1 acts in both checkpoint signaling and replication fork progression (at least in the context of HU stress). ChIP-qPCR experiments confirm localization of Chl1 to stalled forks, and a deficit in cohesin recruitment in chl1Δ. Scc1 overexpression experiments suggest that the role of CHL1 in HU resistance is not due to an Scc1 functional deficit. Therefore, the authors focus on other possible roles for Chl1 in stalled fork recovery. They find that replication is advanced in chl1Δ cells. To me, this mirrors the phenomena discovered by the Pasero and Brown labs, where low constitutive checkpoint activation in a variety of genetic backgrounds results in accumulation of dNTPs, particularly in experiments where cells are arrested in G1. The dNTP accumulation leads, indirectly, to increased replication fork rate. The authors refer to this as regulation, which to me implies something much more direct and active than what is likely to be occurring in chl1Δ. They also find that CHL1 affects RPA association with stalled forks, and therefore impinges on checkpoint activation, and suggest a more direct role for this function. The paper contains a considerable amount of interesting data, and encourages the view that Chl1 function is complex and poorly understood. Although the conclusions are sometimes overstated ('controls' 'regulates'), the data are compelling and clean, and definitely extend our current understanding of Chl1 function. I think the most important of my comments can be addressed by adjustments to the text.

Comments:

1. Genetic interactions of chl1Δ have also been analysed by the Boone lab (including noting a strong genetic interaction with mrc1Δ), and so their work could be cited. There is quite a bit of data here that has been previously published (helicase-dead chl1, Chl1 ChIP to forks), although I can see that most is foundation for subsequent experiments.

2. The ChIP experiments would be stronger if a ChIP-seq approach had been used. In particular, the inference that mrc1Δ and chl1Δ are not additive for cohesin recruitment to stalled forks rests on a single time point at a single origin. The genetic data with respect to additive effects on cohesion in Xu et al were quite clear. Effects could of course be different at stalled forks vs generic chromosome sites, so I don't think the genetic data that suggest CHL1 and MRC1 are parallel for cohesion establishment are

refuted by the data here. So perhaps in the discussion, the point might be that the current data might reveal a specificity of function that was not evident at more general chromosome sites (and Scc1 loading is not exactly the same as cohesion, but is a proxy for it). Please consider using 'super plots' instead of bar graphs in Fig 2, so that all individual data points are shown. And please replace SEM error bars with SD error bars.

3. The experiment in Fig 3, formally speaking, does not assess fork restart or stalled fork recovery. Rather it assesses HU resistance in a viability assay. I would modify the text accordingly.

4. In the *mrc1Δ chl1Δ pol epsilon* ChIP, the interpretation of the data isn't clear. Is pol epsilon absent, or simply elsewhere? Are the authors arguing that replication is more advanced in the double mutant, or is replication progression defective? Combing might address fork rate more clearly than does ChIP. Also, 'controls' and 'regulates' suggest an active and direct effect, whereas the effect here is likely passive and indirect (as the authors acknowledge later in the ms).

5. I didn't find the data reflecting the absence of checkpoint activation in *chl1Δ* to be compelling. Rad52 foci are not a particularly sensitive measure of damage since there is quite a high background in unperturbed cells. The gist of previous analyses was that very low levels of checkpoint activation are enough to cause dNTP accumulation when cells are arrested in G1. It appears in Fig EV6 that RNR3 is upregulated. What is the proposed mechanism by which CHL1 would directly regulate RNR1 transcription? Could the hypothesis that RNR1 is upregulated via the DDR be tested more directly? If RNR1 is truly upregulated in a more direct and DDR independent manner by CHL1, that would be quite unexpected and interesting, and would certainly merit further analysis.

6. The effects of *chl1Δ* on RPA loading and checkpoint activation were quite clear.

Reviewer #3 (Comments to the Authors (Required)):

In this manuscript, Batte and coworkers examine the cause of a previously reported genetic interaction between CHL1 and MRC1. CHL1 helicase is a known component of the eukaryotic replisome that is recruited to replication forks via the adaptor protein Ctf4, and promotes sister chromatid cohesion. The authors outline two pathways by which CHL1 affects genome duplication upon replication stress: regulation of RNR1, which control dNTP levels, and the binding of RPA to stalled replication forks.

Overall, I feel the experiments are thorough and have been well executed. Whilst the data largely support the conclusions drawn and the manuscript seems suitable for publication in Life Science Alliance, there a certain aspects I feel should be altered or addressed prior to publication.

Figure 2. Passage on page 8 'suggesting that cohesin accumulates on chromatin following passage of the replisome.' This should be changed to 'suggesting that at least a proportion of Scc1 may accumulate on chromatin following passage of the replisome'

As the authors note in the next section, they also observe Scc1 at the late origin ARS501, which should not fire in the presence of HU, inconsistent with all Scc1 signal being replication-dependent.

Figure 3. The fitted regression lines in figures 3c-3f do not seem to reflect the undying data points well. This is particularly clear for the *Chl1Δ* sample in Fig. 3e, where there is a peak for Pol Alpha 1 kb from the origin, but this is not reflected in the fitted line. The data, and all such analyses throughout the manuscript, should instead be presented with simple connecting lines between data points.

To make the data easier to visualise, the authors might also consider plotting all time points from one ChIP assay in a single chart, but separating out the WT and *Chl1Δ* samples. For example, combining the WT samples from Figures 3C, EV3A and EV3B on one chart, and the *Chl1Δ* samples from these figures on another chart. This would make it easier to look at how the ChIP signal changes with time, whilst still enabling a comparison between WT and *Chl1Δ*.

Given the data, I feel the conclusion on page 11 'These results indicate that in HU-treated CHL1-deficient cells the faster DNA synthesis is associated with a faster progression of the replication machinery' should be moderated.

Figure EV5F. DNA content in the 60 min time point in the absence of Chl1 seems significantly more than 2C. Could this reflect some form of rereplication taking place, or might the authors have another explanation for this? It should be mentioned in the text.

Figure 5. The authors describe the impact of Chl1 on dNTP levels and the recruitment of RPA to chromatin as distinct pathways. However, these two events could be linked by the fact that increased dNTP levels in the absence of Chl1 could lead to less

stalling of DNA synthesis in the presence of HU, which would be expected to reduce the amount of RPA recruited to chromatin. Have the authors analysed the impact on RPA recruitment of increasing dNTPs via another pathway, for example deleting Sml1? Can they exclude that the impact of Chl1 on RPA recruitment is not due to the dNTP effect?

Page 16/17 states 'These results suggest that the helicase activity of Chl1 directly controls replication fork rate by regulating proper formation of RPA-coated ssDNA once cells progress through S-phase, thereby activating the Rad53-dependent intra-S checkpoint.' As above, can the authors exclude that the effect on fork rate is not entirely due to increased dNTP levels in the absence of Chl1?

Point-by-point response to the reviewer's comments**LSA-2021-01153-T****Chl1 helicase controls replication fork progression by regulating dNTP pools**

Amandine Batté, Sophie C. van der Horst, Mireille Tittel-Elmer, Su Ming Sun, Sushma Sharma, Jolanda van Leeuwen, Andrei Chabes and Haico van Attikum

We would like to thank the referees for their feedback and constructive comments on our manuscript. Based on this we have performed several additional experiments, which were added to the manuscript. The results of these experiments modify our initial findings and model. The effect of Chl1 on dNTP levels seems to be the cause of the other phenotypes we observed. To better highlight this new finding, we changed the order of the paragraphs of the result section and modified the discussion. Below you will find a point-by-point response to the comments (text in red). All changes to the manuscript are also indicated with text in red.

Reviewer #1 (Comments to the Authors (Required)):

Batté and co-workers address the role of the Chl1 (DDX11) helicase in DNA replication fork progression under replication stress conditions in budding yeast. This has recently been studied by Delamarre et al., *Mol Cell* 77, 395-410 (2020), who have shown that the Chl1 helicase activity is required for cohesin loading and nascent DNA resection at stalled DNA replication forks. Thus, Chl1/DDX11 activity contributes to ssDNA formation and RPA association/checkpoint signalling at stalled forks, and ultimately fork restart (Delamarre et al., 2020; Simon et al., *Life Sci Alliance* 3, e201900547 (2020)). This mechanism is distinct from the canonical role of Chl1 in sister chromatid cohesion establishment independently of its helicase activity during unperturbed DNA replication (Samora et al., *Mol Cell* 63, P371-384 (2016)). The results in the current MS by Batté et al. are consistent with the Samora, Delamarre, and Simon studies, and similar conclusions are drawn.

Batté and co-workers have previously detected a negative genetic interaction between chl1 and mrc1 (Ming Sun et al., *J Cell Sci* 133, jcs237628 (2020), see also Costanzo et al., *Science* 353, pii: aaf1420 (2016)), leading to the hypothesis of a contribution of Chl1 to the replication stress response. This genetic interaction is apparent as a growth defect in unperturbed conditions and synthetic hypersensitivity to dNTP depletion induced by HU. The authors show (Fig. 1) that loss of Chl1 negatively synergises with a slow replication and a checkpoint signalling mutant of Mrc1, while growth defects in unperturbed conditions and HU hypersensitivity in complete absence of Mrc1 and WT Chl1 is not rescued by expression of a helicase-dead version of Chl1. Since Chl1's canonical cohesion loading function does not depend on its helicase activity (Samora et al., *Mol Cell* 63, P371-384 (2016)), this implies that growth/replicative defects in mrc1 chl1 double-mutant cells cannot be explained by Chl1's role in cohesion loading in unperturbed conditions. This notion is elaborated further (Fig. 2) by providing evidence through cohesion ChIP experiments that, as expected, Chl1 is required for cohesion loading during replication, but that over-expression of SCC1 (previously shown to overcome the chl1 cohesion loading defect, Samora et al., *Mol Cell* 63, P371-384 (2016)), does not suppress a loss of viability observed for mrc1 chl1 double deletion mutants after a prolonged period of replication arrest. Replication profiling and replication factor ChIP analyses (Fig. 3) are used to show that replication in the presence of HU proceeds faster in chl1 mutant cells, which is consistent with DNA combing experiments performed by the Uhlmann lab (see Fig. 4C in Samora et al., *Mol Cell* 63, P371-384 (2016)) showing directly a statistically significant increase in nascent DNA track length in the presence of HU after 90 min when Chl1 is absent. This may relate in part to elevated G1 dNTP levels in chl1 mutants detected by Batté and co-workers (Fig. 4), which coincides with increased RNR1 expression levels in chl1 deletion or helicase-dead mutants. Finally, a dependence of RPA accumulation at stalled forks on the Chl1 helicase first described by Delamarre and co-workers can be confirmed (Fig. 5), and a reduction in RPA correlates with a reduction in checkpoint activation as detected by Rad53 phosphorylation, consistent with RPA and ssDNA providing the landing pad for the apical kinase Mec1-Ddc2, and also consistent with mammalian data (Simon et al., *Life Sci Alliance* 3, e201900547 (2020)). The authors conclude with a model (Fig. 5F) summing up (1) the role of Chl1 in RPA establishment at stalled DNA replication forks as previously described and (2) add a second aspect, namely a Chl1-

dependent downregulation of RNR1 at S phase onset leading to reduced dNTP levels, and therefore reduced initial replication speed.

Although the current MS overlaps largely with recent reports on the role of Chl1 at stalled DNA replication forks, the work is carried out to a very high standard and provides additional validation of current models (Delamarre et al., 2020; Simon et al., Life Sci Alliance 3, e201900547 (2020)). There are some overstatements in the current version of the MS that should be addressed, and additional evidence is critical to support a regulatory role of the Chl1 helicase on RNR1 expression.

Major points:

-There is probably one aspect in which the current MS goes beyond the current literature on Chl1 functions. This is the potential involvement of the Chl1 helicase activity in RNR1 expression and therefore dNTP levels (see model Fig. 5F). At present the data supporting a transcriptional control function for Chl1 is not very strong and it would be key to provide additional support for the proposed model. The authors point out the caveat that dNTP levels may simply be deregulated in *chl1* mutants due to elevated intrinsic replication stress levels. This is addressed by showing that Rad52 foci are not elevated in *chl1* deletion mutants, but the question remains whether this assay is sensitive enough to detect elevated replication stress and the possibility that dNTP levels may be deregulated because of that. It appears that *chl1* mutants have a cell morphology indicative of replication stress and damage and large dumbbell shaped cells are apparent (Fig. EV6D). Is this concomitant with chronic checkpoint activation? To rule out multi-cell cycle effects caused by a CHL1 deletion, it may be possible to use a degradation tag and acutely deplete Chl1, perhaps in G1, and monitor effects on RNR1 expression. An alternative approach could be to experimentally titrate Chl1 levels and assess how RNR1 expression responds. While this would not mechanistically resolve how Chl1 helicase activity may play a role in RNR1 gene expression, these experiments, or perhaps other supporting data the authors may be able to provide, would be helpful to strengthen the Chl1-RNR1 link proposed in the MS.

We did not observe any chronic checkpoint activation by measuring Rad53 phosphorylation in G1-arrested *chl1Δ* cells (see Fig 3D; t0). Moreover, the formation of spontaneous Rad52 foci was unaffected in *chl1Δ* cells (see Fig EV6A and B). Finally, we also did not observe an increase in the expression of the DNA damage responsive *HUG1* gene in *chl1Δ* cells (see Fig EV6C). Taken together, the results from these three independent approaches indicate that replication stress and/or DNA damage levels are not elevated in *chl1Δ* cells.

To rule out any effect of chronic loss of CHL1 from previous cell cycles, we measured the expression of the four *RNR* genes in G1-arrested Chl1-AID-ostTIR1 cells that were pre-treated with auxin for 5 hours to deplete AID degraon-tagged Chl1 (see new Figs EV7E-F). We observed an increase in *RNR1* expression in the G1-arrested and auxin-treated Chl1-AID-ostTIR1 cells, which was almost comparable to that observed in *chl1Δ* cells (see Fig 6F), suggesting that expression of *RNR1* is due to the acute depletion of Chl1 and does not arise from chronic replication stress and/or DNA damage. The new data are presented in Fig. EV7E-F and discussed on page 17 of the revised manuscript.

To further explore the Chl1-RNR1 link, we have performed ChIP experiments to assess whether Chl1-Myc would bind to the *RNR1*, *RNR2*, *RNR3* or *RNR4* promoter in G1-arrested cells. However, we did not detect any enrichment for Chl1 at these loci, while Chl1 was enriched near ARS607 in HU-treated S-phase cells (see new Fig. EV7H), agreeing with our previous observations (Fig 2B). Alternatively, we assessed whether Chl1 might regulate *RNR1* transcription by controlling *SML1* expression. However, we did not observe any impact of CHL1 loss on *SML1* expression (see new Fig EV7G). Altogether, these data indicate that the Chl1 helicase does not directly regulate *RNR1* or *SML1* transcription. The mechanism by which Chl1 regulates *RNR* expression remains elusive, but elucidating it is a project on its own and therefore beyond the scope of the current study. The new data are presented in Fig EV7G and H and discussed on pages 17-18 of the revised manuscript.

-Key points in the discussion are similar to those in previous work, in particular in the Delamarre 2020 paper mentioned above. A clear description of the state-of-the art of Chl1 at stalled replication forks, summarizing the key papers (Delamarre et al., 2020; Simon et al., Life Sci Alliance 3, e201900547 (2020); Samora et al., Mol Cell 63, P371-384 (2016)) might be very helpful for readers. This would allow to clearly state which aspects of the work are of a more confirmatory nature, and to then focus on more novel aspects, i.e., the potential link between Chl1 helicase activity and RNR1 expression and the benefits this mechanism might have.

The discussion was profoundly rewritten to summarize the previous studies and better highlight the novel aspects of our paper.

Minor points:

-The experiment shown in Fig. 1D should be carried out with the *mrc1* separation-of-function mutants used in 1C to support the statement at the bottom of p. 7 that "Chl1's helicase activity is required for checkpoint control and replication fork progression."

We have integrated empty vector (EV), *CHL1* or *chl1*^{K48R} expression cassettes at the *HIS3* locus in *chl1Δ mrc1*^{AQ} and *chl1Δ mrc1*¹⁻⁸⁴³ strains and repeated the drop test on HU (see new Fig 1E). While *CHL1* was able to rescue the fitness defect of both *chl1Δ mrc1*^{AQ} and *chl1Δ mrc1*¹⁻⁸⁴³ cells, *chl1*^{K48R} could not. These results support the notion that the helicase function of Chl1 is required for checkpoint control and replication fork progression. The new data are presented in Fig 1E and discussed on page 7 of the revised manuscript.

-Could the authors clarify how the experiment in Fig. 2F showing that prolonged exposure of cells to 200 mM HU results in a significant loss of viability in the absence of Mrc1 or Chl1/Mrc1 (there is only little effect on viability when only Chl1 is absent under the conditions employed), and that this effect is not suppressed by overexpression of Scc1, warrants the conclusion on p. 10 that "cohesin loading mediated by Chl1 is not involved in the recovery from replication fork stalling"? On face value it appears that Scc1 over-expression under these conditions is not sufficient to promote DNA replication fork stability and/or restart when Mrc1 is absent.

We showed in Fig 2D and Fig EV1L that Mrc1 does not affect cohesin loading at stalled replication forks. It has been previously demonstrated that Mrc1 is rather necessary for the stabilization of stalled forks (Katou et al., 2003; Szyjka et al., 2005, Tourrière et al., 2005). Consistent with the observation that Mrc1 loss does not lead to cohesin defects, we found that Scc1 over-expression does not overcome the replication fork stability and/or restart defects observed in the absence of Mrc1. A sentence explaining these observations was added on page 10 of the revised manuscript. In addition, both *chl1Δ* and *chl1Δ mrc1Δ* showed a decrease in cohesin loading and loss of viability after prolonged exposure to HU. However, restoring the cohesin loading defect through Scc1 overexpression did not rescue the viability of these two strains, while it did for *scc1-73* and *chl1Δ scc1-73*, indicating that Chl1's function in cohesin loading is dispensable for the recovery from replication fork stalling.

-The statement on p. 10 with regard to replication profiling that "no difference was observed 90 minutes after release from HU, in accordance with a previous study (Samora et al., 2016)" is slightly misleading. The Samora study clearly shows that replication in the presence of HU proceeds faster when Chl1 is absent at the 90 min time-point after release of synchronized cells into 200 mM HU (see Fig. 4C in Samora et al., Mol Cell 63, P371-384 (2016)). Having said this, the trends in the Samora study and the current MS are similar. Perhaps this can be rephrased to avoid confusion.

The sentence on page 10 has been rephrased to: "*However, no difference was observed 90 minutes after release in HU, which suggests an increase in the rate of DNA synthesis in CHL1-deficient cells early after release in S-phase in the presence of HU. This is in accordance with a previous study that showed that DNA replication was faster at early time points S-phase in chl1Δ compared to WT (Samora et al., 2016).*"

Reviewer #2 (Comments to the Authors (Required)):

Batte et al examine the role of the DNA helicase Chl1 in the DNA replication stress response. Chl1 has previously been connected to sister chromatid cohesion, in a pathway parallel to the checkpoint mediator Mrc1. The authors start from the observation that *chl1Δ* and *mrc1Δ* have a strong negative genetic interaction, following up with separation of function alleles to suggest that *CHL1* acts in both checkpoint signaling and replication fork progression (at least in the context of HU stress). CHIP-qPCR experiments confirm localization of Chl1 to stalled forks, and a deficit in cohesin recruitment in *chl1Δ*. Scc1 overexpression experiments suggest that the role of *CHL1* in HU resistance is not due to an Scc1 functional deficit. Therefore, the authors focus on other possible roles for Chl1 in stalled fork recovery. They find that replication is advanced in *chl1Δ* cells. To me, this mirrors the phenomena discovered by the Pasero and Brown labs, where low constitutive checkpoint activation in a variety of genetic backgrounds results in accumulation of dNTPs, particularly in experiments where cells are arrested in

G1. The dNTP accumulation leads, indirectly, to increased replication fork rate. The authors refer to this as regulation, which to me implies something much more direct and active than what is likely to be occurring in *chl1Δ*. They also find that CHL1 affects RPA association with stalled forks, and therefore impinges on checkpoint activation, and suggest a more direct role for this function. The paper contains a considerable amount of interesting data, and encourages the view that Chl1 function is complex and poorly understood. Although the conclusions are sometimes overstated ('controls' 'regulates'), the data are compelling and clean, and definitely extend our current understanding of Chl1 function. I think the most important of my comments can be addressed by adjustments to the text.

Comments:

1. Genetic interactions of *chl1Δ* have also been analysed by the Boone lab (including noting a strong genetic interaction with *mrc1Δ*), and so their work could be cited. There is quite a bit of data here that has been previously published (helicase-dead *chl1*, Chl1 ChIP to forks), although I can see that most is foundation for subsequent experiments.

Citations to Costanzo et al, *Science* (2016) and Xu et al., *Genetics* (2007) were added on page 6 of the revised manuscript.

2. The ChIP experiments would be stronger if a ChIP-seq approach had been used. In particular, the inference that *mrc1Δ* and *chl1Δ* are not additive for cohesin recruitment to stalled forks rests on a single time point at a single origin. The genetic data with respect to additive effects on cohesion in Xu et al were quite clear. Effects could of course be different at stalled forks vs generic chromosome sites, so I don't think the genetic data that suggest CHL1 and MRC1 are parallel for cohesion establishment are refuted by the data here. So perhaps in the discussion, the point might be that the current data might reveal a specificity of function that was not evident at more general chromosome sites (and *Sccl* loading is not exactly the same as cohesion, but is a proxy for it). Please consider using 'super plots' instead of bar graphs in Fig 2, so that all individual data points are shown. And please replace SEM error bars with SD error bars.

ChIP experiments performed with WT, *chl1Δ*, *mrc1Δ* and *chl1Δ mrc1Δ* did not only assess Chl1-Myc binding near ARS607, but also monitored its binding near ARS305 in G1 and HU-treated S-phase cells (Fig2D and Fig EV1L). Since cohesin loading was not further affected in *chl1Δ mrc1Δ* compared to each single mutant at both ARSs, we concluded that *mrc1Δ* and *chl1Δ* are not additive for cohesin recruitment to stalled replication forks. The *Sccl* ChIPs that were performed over time in HU (Fig 2C and Fig EV1J) showed that cohesin was fully recruited to the ARSs 40 min after release from G1, a time point at which we could detect a clear defect in cohesin recruitment in *chl1Δ* compared to WT. The difference in *Sccl* binding at 60 min compared to 40 min was minor. For this reason, we decided to conduct the experiment in the *chl1Δ mrc1Δ* at a single time point (40 min) showing high levels of *Sccl* recruitment to these ARSs.

To better differentiate between the role of *Mrc1* and *Chl1* in cohesion establishment at chromosomal loci and cohesin loading at stalled forks, we added the following statement page 20 of the revised manuscript: "*This also suggests that the additive function of Mrc1 and Chl1 in cohesion establishment at loci across the genome may differ from cohesin loading at stalled replication forks.*"

All bar graphs were changed to super plots showing individual data points as requested by the reviewer. However, we kept SEM error bars as it is considered to be more appropriate to show inferential error bars (such as SE or CI) rather than SD when comparing experimental results with controls (Cumming et al., *JCB*, 2007).

3. The experiment in Fig 3, formally speaking, does not assess fork restart or stalled fork recovery. Rather it assesses HU resistance in a viability assay. I would modify the text accordingly.

The reviewer may refer to Fig 2F instead of Fig 3. We have changed the statement on page 9 "*we examined whether overexpression of Scc1 could rescue the ability of chl1Δ, mrc1Δ and chl1Δ mrc1Δ to recover from HU-induced replication fork stalling*" to "*we examined whether overexpression of Scc1 could rescue the viability of chl1Δ, mrc1Δ and chl1Δ mrc1Δ following exposure to HU*".

4. In the *mrc1Δ chl1Δ pol epsilon* ChIP, the interpretation of the data isn't clear. Is *pol epsilon* absent, or simply elsewhere? Are the authors arguing that replication is more advanced in the double mutant, or is replication progression defective? Combing might address fork rate more clearly than does ChIP.

Also, 'controls' and 'regulates' suggest an active and direct effect, whereas the effect here is likely passive and indirect (as the authors acknowledge later in the ms).

We meant to say that replication fork progression is defective in *chl1Δ mrc1Δ*. This is supported by the replication timing analysis (Fig 4I) showing that replication cannot be completed in *chl1Δ mrc1Δ* within the 60 minutes window of HU exposure, during which WT and single mutants can do so. The conclusion that replication fork progression is defective *chl1Δ mrc1Δ* has been added on page 14 of the revised manuscript.

Measurements of BrdU tracks/DNA combing after short pulses are not reliable due to the resolution limits of this technique. Indeed, a 3 kb track is seen as a single dot (Pasero *et al.*, Genes Dev, 2002) and a comparison with electron microscopy data showed that short tracks measured by DNA combing are largely overestimated in size (Giannattasio *et al.*, Mol Cell, 2010). Therefore, the analysis of replication kinetics by DNA combing at early time points and a comparison to our replication timing/ChIP assays is not feasible.

5. I didn't find the data reflecting the absence of checkpoint activation in *chl1Δ* to be compelling. Rad52 foci are not a particularly sensitive measure of damage since there is quite a high background in unperturbed cells. The gist of previous analyses was that very low levels of checkpoint activation are enough to cause dNTP accumulation when cells are arrested in G1. It appears in Fig EV6 that RNR3 is upregulated. What is the proposed mechanism by which CHL1 would directly regulate RNR1 transcription? Could the hypothesis that RNR1 is upregulated via the DDR be tested more directly? If RNR1 is truly upregulated in a more direct and DDR independent manner by CHL1, that would be quite unexpected and interesting, and would certainly merit further analysis.

We did not observe any chronic checkpoint activation by measuring Rad53 phosphorylation in G1-arrested *chl1Δ* cells (see Fig 3D; t0). Moreover, the formation of spontaneous Rad52 foci was unaffected in *chl1Δ* cells (see Fig EV6A and B). Finally, we also did not observe an increase in the expression of the DNA damage responsive *HUG1* gene in *chl1Δ* cells (see new Fig EV6C). Taken together, the results from these three independent approaches indicate that replication stress and/or DNA damage levels are not elevated in *chl1Δ* cells.

The fact that there is no increase in spontaneous Rad52 foci, Rad53 activation or *HUG1* expression suggests no increase in spontaneous DNA damage/replication stress that could have triggered a DNA damage response in *chl1Δ* mutants that suffered from long-term depletion of Chl1. In support of this we now found that short-term depletion of AID degraon-tagged Chl1 in auxin-treated Chl1-AID-ostIR1 strains already leads to increased *RNR1* expression (but not *RNR2*, *RNR3* or *RNR4*) (see reviewer 1, point 1 and Fig EV7E-F). Thus, it is unlikely that *RNR* gene expression is regulated in a DNA damage response-dependent manner in the absence of Chl1.

To explore whether Chl1 may directly regulate *RNR* expression, we have performed ChIP experiments to assess whether Chl1-Myc would bind to the *RNR1*, *RNR2*, *RNR3* or *RNR4* promoter in G1-arrested cells. However, we did not detect any enrichment for Chl1 at these loci, while Chl1 was enriched near ARS607 in HU-treated S-phase cells (see new Fig. EV7H, agreeing with our previous observations (Fig 2B). Alternatively, we assessed whether Chl1 might regulate *RNR1* transcription by controlling *SML1* expression. However, we did not observe any impact of *CHL1* loss on *SML1* expression (see new Fig EV7G). Altogether, these data indicate that the Chl1 helicase does not directly regulate *RNR1* or *SML1* transcription. The mechanism by which Chl1 regulates *RNR* expression remains elusive, but elucidating it is a project on its own and therefore beyond the scope of the current study. The new data are presented in Fig EV7G and H and discussed on pages 17-18 of the revised manuscript.

6. The effects of *chl1Δ* on RPA loading and checkpoint activation were quite clear.

We thank the reviewer for this positive feedback.

Reviewer #3 (Comments to the Authors (Required)):

In this manuscript, Batte and coworkers examine the cause of a previously reported genetic interaction between CHL1 and MRC1. CHL1 helicase is a known component of the eukaryotic replisome that is recruited to replication forks via the adaptor protein Ctf4, and promotes sister chromatid cohesion. The authors outline two pathways by which CHL1 affects genome duplication upon replication stress: regulation of RNR1, which control dNTP levels, and the binding of RPA to stalled replication forks.

Overall, I feel the experiments are thorough and have been well executed. Whilst the data largely support the conclusions drawn and the manuscript seems suitable for publication in Life Science Alliance, there are certain aspects I feel should be altered or addressed prior to publication.

Figure 2. Passage on page 8 'suggesting that cohesin accumulates on chromatin following passage of the replisome.' This should be changed to 'suggesting that at least a proportion of Scc1 may accumulate on chromatin following passage of the replisome'

As the authors note in the next section, they also observe Scc1 at the late origin ARS501, which should not fire in the presence of HU, inconsistent with all Scc1 signal being replication-dependent.

We never intended to suggest that all Scc1 signal is replication-dependent as we indeed observed that Scc1 was also present at the late origin ARS501 (Fig 2C), as well as at several other chromosomal loci (Fig EV1K). However, we agree that the passage on page 8 was misleading and we changed the sentence according to the reviewer's suggestion.

Figure 3. The fitted regression lines in figures 3c-3f do not seem to reflect the underlying data points well. This is particularly clear for the Chl1Δ sample in Fig. 3e, where there is a peak for Pol Alpha 1 kb from the origin, but this is not reflected in the fitted line. The data, and all such analyses throughout the manuscript, should instead be presented with simple connecting lines between data points.

We changed all fitted lines to simple connecting lines throughout the manuscript as suggested by the reviewer.

To make the data easier to visualise, the authors might also consider plotting all time points from one ChIP assay in a single chart, but separating out the WT and Chl1Δ samples. For example, combining the WT samples from Figures 3C, EV3A and EV3B on one chart, and the Chl1Δ samples from these figures on another chart. This would make it easier to look at how the ChIP signal changes with time, whilst still enabling a comparison between WT and Chl1.

We originally presented the data as suggested by the reviewer. However, that way the comparison between WT and *chl1Δ* was difficult to comprehend. In addition, because the main point was to compare how replisome recruitment is affected in *chl1Δ*, we decided to keep the actual representation, which allows a better visualization of these differences in our view.

Given the data, I feel the conclusion on page 11 'These results indicate that in HU-treated CHL1-deficient cells the faster DNA synthesis is associated with a faster progression of the replication machinery' should be moderated.

We modified the conclusion to "*These results indicate that in HU-treated CHL1-deficient cells the faster DNA synthesis is accompanied by a faster progression of the replication machinery*" as suggested by the reviewer.

Figure EV5F. DNA content in the 60 min time point in the absence of Chl1 seems significantly more than 2C. Could this reflect some form of rereplication taking place, or might the authors have another explanation for this? It should be mentioned in the text.

We added a sentence on page 14 of the revised manuscript to highlight the more than 2C DNA content in *chl1Δ* cells after 60 minutes into HU, and indicate that this may reflect endoreplication in these cells.

Figure 5. The authors describe the impact of Chl1 on dNTP levels and the recruitment of RPA to chromatin as distinct pathways. However, these two events could be linked by the fact that increased dNTP levels in the absence of Chl1 could lead to less stalling of DNA synthesis in the presence of HU, which would be expected to reduce the amount of RPA recruited to chromatin. Have the authors analysed the impact on RPA recruitment of increasing dNTPs via another pathway, for example deleting *Sml1*? Can they exclude that the impact of Chl1 on RPA recruitment is not due to the dNTP effect?

As suggested by the reviewer, we have performed ChIP experiments for RPA in WT, *chl1Δ* and *sml1Δ* strains. As expected, we observed reduced RPA levels near ARS607 in *chl1Δ* HU-treated S-phase

cells. Strikingly, we also found that loss of *SML1* severely reduces the recruitment of RPA near ARS607 in HU-treated S-phase cells (see new Fig EV7A). This suggests that the increase in dNTPs levels observed in *chl1Δ* and *sml1Δ* leads to reduced RPA levels at replication forks. This in turn may prevent checkpoint-dependent replication fork stalling, explaining the increase in fork progression observed in the absence of Chl1 (Fig 4B-E). The impact of Chl1 on RPA recruitment and fork progression may therefore not be directly related to its helicase function as we initially suggested, but rather reflect its impact on dNTP production. The new data are presented in Fig EV7A and discussed on page 16 of the revised manuscript.

Page 16/17 states 'These results suggest that the helicase activity of Chl1 directly controls replication fork rate by regulating proper formation of RPA-coated ssDNA once cells progress through S-phase, thereby activating the Rad53-dependent intra-S checkpoint.' As above, can the authors exclude that the effect on fork rate is not entirely due to increased dNTP levels in the absence of Chl1?

We previously observed that the fork rate was higher in *chl1Δ sml1Δ* compared to *sml1Δ* (Fig 6D and E). Thus, we measured dNTP levels in WT, *chl1Δ*, *sml1Δ*, *chl1Δ sml1Δ* in G1-phase and in S-phase after 60 min in HU to determine whether increased dNTPs levels in the absence of Chl1 directly affects fork rate. We found that the loss of both *SML1* and *CHL1* has an additive effect on the dNTP levels compared to that in each single mutant. This suggests that the effect on fork rate is entirely due to the increased dNTP levels in the absence of Chl1 and not a direct effect of its helicase function as we initially suggested. The new data are added to Fig 6A and discussed on page 16 of the revised manuscript.

December 20, 2021

RE: Life Science Alliance Manuscript #LSA-2021-01153-TR

Prof. Haico van Attikum
Leiden University Medical Center
Human Genetics
Einthovenweg 20
Leiden 2333ZC
Netherlands

Dear Dr. van Attikum,

Thank you for submitting your revised manuscript entitled "Chl1 helicase controls replication fork progression by regulating dNTP pools". We would be happy to publish your paper in Life Science Alliance pending final revisions necessary to meet our formatting guidelines.

- As requested by reviewer 3, please remove the word 'directly' to the sentence on page 11 ('Taken together, these results suggest that Chl1's helicase activity is directly involved in the accumulation of ssDNA and the proper loading of RPA at stalled replication forks') as the new data support an indirect regulation of RPA accumulation through dNTP levels
- please add ORCID ID for secondary corresponding author-they should have received instructions on how to do so
- please add your main, supplementary figure, and table legends to the main manuscript text after the references section;
- LSA allows supplementary figures, but no EV Figures; please update your callouts for the Supplementary Figures in the manuscript Fig EV1A=Fig S1A; while supplementary figures use the system supplementary Fig S1;
- we encourage you to revise the figure legends for figures S1, S7 such that the figure panels are introduced in an alphabetical order
- please indicate molecular weight next to each protein blot
- please upload your Tables in editable .doc or excel format
- please add callouts for Figures S3A-G and S4A-E to your main manuscript text;
- please provide a separate Data Availability section if applicable

FIGURE CHECKS:

- please provide scale bar or objective magnification for figure S6A

A. FINAL FILES:

- An editable version of the final text (.DOC or .DOCX) is needed for copyediting (no PDFs).
- High-resolution figure, supplementary figure and video files uploaded as individual files: See our detailed guidelines for preparing your production-ready images, <https://www.life-science-alliance.org/authors>
- Summary blurb (enter in submission system): A short text summarizing in a single sentence the study (max. 200 characters)

including spaces). This text is used in conjunction with the titles of papers, hence should be informative and complementary to the title. It should describe the context and significance of the findings for a general readership; it should be written in the present tense and refer to the work in the third person. Author names should not be mentioned.

B. MANUSCRIPT ORGANIZATION AND FORMATTING:

Sincerely,

Reviewer #1 (Comments to the Authors (Required)):

In their revised MS, the authors have added substantial new experimental data and have addressed all the points raised during peer review. Discussion of previously published work by other labs has been integrated appropriately. The notion of Chl1 having a mechanistically unresolved impact on RNR1 expression and dNTP levels has been strengthened, while a clear indication in the model that this is likely to be indirect and requires further investigation appropriately acknowledges the current uncertainties on this point. I support publication of the MS in its revised form.

Reviewer #2 (Comments to the Authors (Required)):

My comments have all been addressed. Thanks.

Reviewer #3 (Comments to the Authors (Required)):

I feel the revisions have significantly improved the manuscript, which now seems suitable for publication in Life Science Alliance.

Minor point:

The authors state on page 11 'Taken together, these results suggest that Chl1's helicase activity is directly involved in the accumulation of ssDNA and the proper loading of RPA at stalled replication forks.'

Given the new data supporting an indirect regulation of RPA accumulation through dNTP levels, the word 'directly' should be

removed here.

December 28, 2021

RE: Life Science Alliance Manuscript #LSA-2021-01153-TRR

Prof. Haico van Attikum
Leiden University Medical Center
Human Genetics
Einthovenweg 20
Leiden 2333ZC
Netherlands

Dear Dr. van Attikum,

Thank you for submitting your Research Article entitled "ChI1 helicase controls replication fork progression by regulating dNTP pools". It is a pleasure to let you know that your manuscript is now accepted for publication in Life Science Alliance. Congratulations on this interesting work.

DISTRIBUTION OF MATERIALS:

Again, congratulations on a very nice paper. I hope you found the review process to be constructive and are pleased with how the manuscript was handled editorially. We look forward to future exciting submissions from your lab.

Sincerely,
